# Cardiac health assessment across scenarios and devices using a multimodal foundation model pretrained on data from 1.7 million individuals

Xiao Gu [1] ✉, Wei Tang[1,2,3], Jinpei Han [4], Veer Sangha[1], Fenglin Liu [1], Shreyank N. Gowda[5], Antonio H. Ribeiro [6], Patrick Schwab [7], Kim Branson[7], Lei Clifton[1,8], Antonio Luiz P. Ribeiro[9], Zhangdaihong Liu [1,10] ✉ & David A. Clifton [1,10] ✉

Cardiovascular diseases remain a major contributor to the global burden of healthcare, highlighting the importance of accurate and scalable methods for cardiac monitoring. Cardiac biosignals, most notably electrocardiograms (ECG) and photoplethysmograms, are essential for diagnosing, preventing and managing cardiovascular conditions across clinical and home settings. However, their acquisition varies substantially across scenarios and devices, whereas existing analytical models often rely on homogeneous datasets and static bespoke models, limiting their robustness and generalizability in diverse real-world contexts. Here we present a cardiac sensing foundation model (CSFM) that leverages transformer architectures and a generative masked pretraining strategy to learn unified representations from heterogeneous health records. CSFM is pretrained on a multimodal integration of data from various large-scale datasets, comprising cardiac signals from approximately 1.7 million individuals and their corresponding clinical or machine-generated text reports. The embeddings derived from CSFM act as effective, transferable features across diverse cardiac sensing scenarios, supporting a seamless adaptation to the varied input configurations and sensor modalities. Extensive evaluations across diagnostic tasks, demographic recognition, vital sign measurement, clinical outcome prediction and ECG question answering demonstrate that CSFM consistently outperforms traditional one-modal-one-task approaches. Notably, CSFM maintains favourable performance across both 12-lead and single-lead ECGs, as well as in scenarios involving ECG only, photoplethysmogram only or a combination of both. This highlights its potential as a versatile and scalable foundation for comprehensive cardiac monitoring.

Cardiovascular diseases are among the leading causes of morbidity and mortality worldwide, underscoring the need for accurate and timely diagnostic methods[1]. In clinical practice, cardiac biosignals, most notably electrocardiograms (ECGs) and photoplethysmograms (PPGs), serve as critical tools for diagnosing, preventing and managing these conditions[2]. ECGs capture the electrical impulses generated by the heart, providing essential information on cardiac rhythm and conduction pathways. By contrast, PPGs track fluctuations in blood volume through optical sensors, enabling the non-invasive monitoring of peripheral blood flow and cardiac output. The synergistic integration of these biosignals holds substantial potential for digital health innovation, with applications ranging from acute clinical settings[3] to continuous home-based care[4].

Advances in sensing technologies have dramatically reshaped the acquisition landscape of cardiac biosignals[5], as illustrated in Fig. 1. In standard hospital settings, comprehensive 12-lead ECGs are routinely used to capture detailed cardiac activity. These recordings are usually supplemented with contextual clinical annotations, either from cardiologists or generated automatically by software, to enhance their diagnostic and prognostic value. In intensive care units (ICUs), a more streamlined approach is adopted, with fewer-lead ECGs often paired with PPG signals to facilitate real-time monitoring and early detection of adverse events. In addition, in ICU step-down wards as well as in-home and community settings, wearable devices such as smart wrist-worn sensors or patches are increasingly used to capture ECG or PPG signals for continuous monitoring.

However, traditional analytical methods often lack the scalability to coordinate data acquired from such diverse devices and environments. Conventional approaches typically require bespoke models tailored to specific signal types, sensing modalities and clinical tasks. These models are frequently developed from scratch and rely on large-scale, annotated datasets with consistent data formats (for example, identical ECG channel configurations or uniform signal types). In clinical practice, such large-scale datasets are often unavailable owing to the inherent heterogeneity of cardiac biosignals and the specialized expertise required for accurate annotation. Consequently, these methods often yield suboptimal performance on a single limited dataset, given the lack of access to sufficiently comprehensive and uniform data.

Furthermore, models developed on fragmented datasets often exhibit limited transferability and may not be directly applicable across diverse healthcare environments. For instance, a model trained on data from routinely collected 12-lead ECGs may fail to generalize to settings in low- and middle-income countries, where portable or wearable devices (for example, wearable ECG/PPG) provide more affordable solutions. Traditional methods, for example, those based on convolutional networks, are typically channel-dependent[6,7], necessitating modifications to the network architecture to accommodate varying channel configurations. This disparity not only highlights substantial inequities in access to state-of-the-art analytical tools but also underscores the urgent need for versatile, scalable solutions capable of robust performance across diverse clinical contexts.

By contrast, in the field of deep learning, there is an emerging trend towards developing foundation models that can derive generic representations through self-supervised training on large-scale datasets[8]. Remarkable achievements have been realized in both natural language processing[9,10] and computer vision[11,12]. However, in the realm of cardiac biosignals, existing foundation models are predominantly based on ECG data and are largely confined to standard 12-lead configurations[13,14]. This restriction to consistent data dimensionalities limits their utility in broader clinical contexts, where diverse sensing modalities and heterogeneous data formats are common.

To address these challenges, we develop a foundation model, the cardiac sensing foundation model (CSFM; Fig. 1), by incorporating heterogeneous data types (including ECGs from various clinical settings, PPG signals and accompanying clinical annotations) to enable robust, scalable performance across diverse healthcare environments. We train our model using advanced transformer architectures, originally developed for natural language processing and renowned for their ability to process sequential data and capture intricate dependencies[15,16]. This sequential processing capability enables CSFM to effectively manage and integrate multimodal, multichannel information, making it particularly well-suited for analysing cardiac biosignals. We use masked training strategies, where signals are partially obscured across temporal and channel dimensions during pretraining, to facilitate pretraining on heterogeneous data inputs, which is a critical approach for aggregating cardiac sensing-related data from diverse sources.

The CSFM is pretrained on a multimodal integration of cardiac biosignals and associated cardiologist descriptions collected from approximately 1.7 million individuals. We systematically evaluate this framework on datasets gathered from different scenarios and devices, demonstrating competitive performance in varied scenarios, including demographic information analysis, cardiovascular disease classification, vital sign measurement, clinical outcome prediction and ECG-based question answering. Unlike traditional biosignal analysis models that are typically specialized for specific tasks or data types, CSFM can learn generalized representations and adapt to a wide range of downstream applications, offering a versatile and scalable tool for comprehensive cardiac biosignal analysis.

## Results

### Pretraining on vast and heterogeneous cardiac health records
The CSFM leverages a generative pretraining approach, masked modelling, to learn generic representations from diverse biomedical signals. The pretraining utilized data from multiple large-scale datasets, including MIMIC-III-WDB[17], MIMIC-IV-ECG[18] and a privately-held large-scale CODE dataset[19,20], as shown in Fig. 1b. To enhance the diversity and comprehensiveness of our pretraining data, we integrated cardiac biosignals with both machine-generated and clinical notes where available. Specifically, we linked MIMIC-III-WDB ECG/PPG data with corresponding ECG reports extracted from MIMIC-III clinical database[17] and for MIMIC-IV-ECG, we associated ECG signals with machine-generated reports. Further statistical details are available in Fig. 1b and Supplementary Information section 1. On the basis of this integration, we applied a masking strategy to obscure channel-wise and temporal-wise information during the pretraining, enabling effective generalization across varied input configurations. To accommodate different computational and deployment needs, we developed three versions of CSFM: CSFM-Tiny, CSFM-Base and CSFM-Large, corresponding to tiny, base and large in terms of parameter count, respectively.

### Downstream evaluation across diverse cardiac sensing scenarios and devices
The objective of CSFM is to promote generalization and flexible deployment across diverse sensing devices and healthcare scenarios, enabling a seamless integration in real-world settings.

The downstream tasks address various clinical applications of physiological waveforms, encompassing a wide range of healthcare needs. These include demographic information analysis, cardiovascular disease classification (PTB-XL[21], CinC17[22], SimBand[23]), vital sign measurement (VitalDB[24]), clinical outcome prediction (VTaC[25], CODE-15[26]) and question answering (ECG-QA[27]). Among these, demographic information recognition, such as gender, body mass index (BMI) and age, is used to uncover basic biological information encoded in cardiac biosignals. The cardiovascular disease classification supports the diagnosis of cardiac conditions by leveraging the rich information in waveforms. The vital sign measurement enables the continuous monitoring of key physiological parameters (for example, blood pressure), whereas clinical outcome prediction aids in risk stratification and long-term patient management. Moreover, question-answering

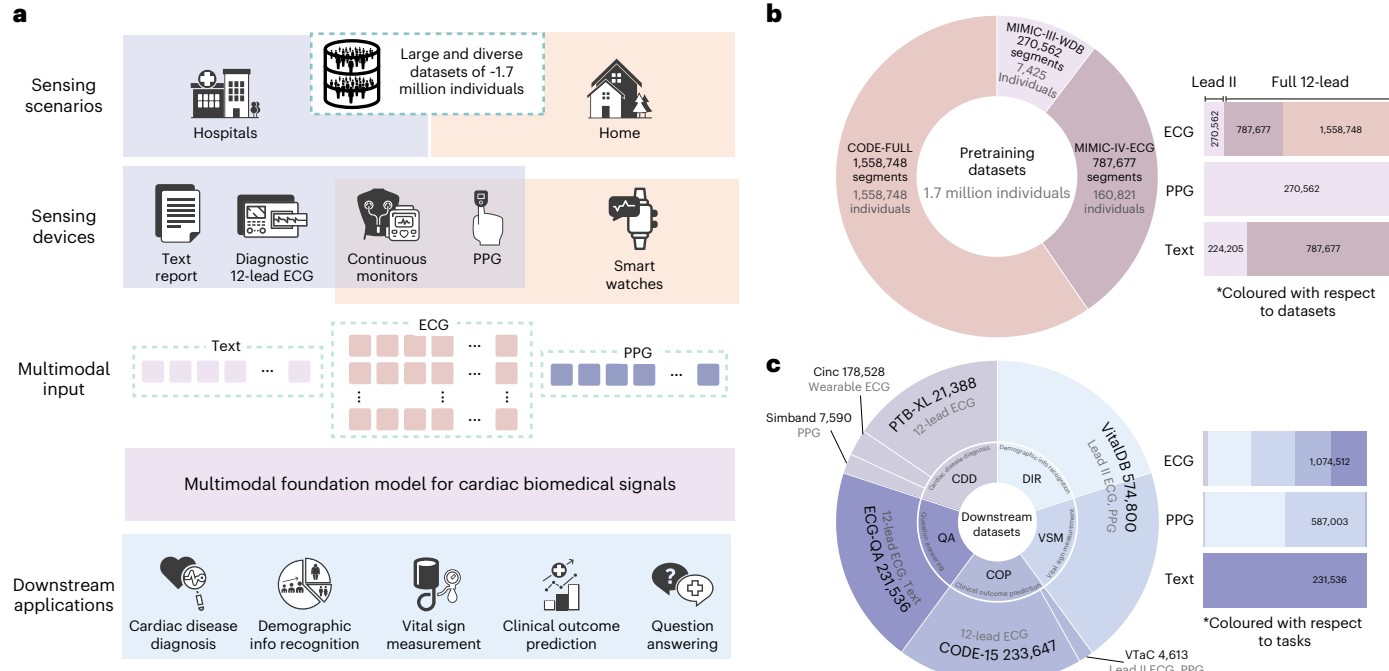

**Fig. 1 | Overview of the CSFM. a**, The CSFM functions as a versatile and scalable framework that learns unified representations from heterogeneous cardiac monitoring data collected from home to hospital settings, including ECG, PPG and clinical text, and supports a wide spectrum of downstream cardiovascular tasks. **b**, The pretraining integrates data from MIMIC-III-WDB (USA), MIMIC-IV-ECG (USA) and CODE-FULL (Brazil), comprising approximately 1.7 million heterogeneous cardiac-related biosignals and texts. Distributions by dataset source and signal modality are shown. **c**, The downstream evaluation covers five representative tasks, including cardiac disease diagnosis (CDD), demographic information recognition (DIR), vital sign measurement (VSM), clinical outcome prediction (COP) and ECG-based question answering (QA), using datasets such as CinC17 (USA), PTB-XL (Germany), SimBand (USA), VTaC (USA), CODE-15 (Brazil) and VitalDB (South Korea), spanning diverse healthcare settings and populations. Detailed dataset descriptions and statistics are provided in Supplementary Information section 1.

tasks integrate both cardiac waveforms and textual queries to provide interpretive insights and enhance decision-making.

Among these tasks, vital sign measurement was treated as a dense regression task, where convolution layers are further added on top of CSFM to perform dense prediction, similar to Ranftl et al.[28]. For all other tasks, we added an additional fully connected layer to perform classification or univariate regression tasks. Please refer to Methods for further details. All downstream comparisons were performed using three random seeds, with mean reported. Where applicable, error bars in visualizations indicate the standard deviation across runs.

The downstream datasets are collected from diverse real-world scenarios, ensuring the comprehensive evaluation of CSFM's adaptability and generalizability, with statistics shown in Fig. 1c and further details in Supplementary Information section 1. Macro-F1 was reported for the multiclass and multilabel task, area under the receiver operating characteristic curve (AUC) for binary classification and mean absolute error (MAE) for regression, with details provided in Supplementary Information section 1.4.

### CSFM generalizes across different healthcare tasks

Traditional biosignal models are typically designed and deployed with fixed input dimensionalities and prediction tasks, making direct transfer across scenarios intractable without large-scale training. CSFM is expected broadly and flexibly adaptable to diverse healthcare tasks, as a versatile tool for comprehensive biosignal analysis.

To highlight the flexibility and generalizability of CSFM, we compared its fine-tuning performance with that of the compared models trained from scratch. This set-up reflects the practical challenges of adopting ECG/PPG models in real-world applications. Figure 2 compared CSFM with multiple basic/advanced deep learning models for medical time series (especially ECG) trained from scratch. These include classification/regression-based models (ResNet1d-18/34/50/101, Inception1D[29], a Multi-Scale Deep Neural Network (MSDNN)[30]) and dense sequence-to-sequence regression models (BiLSTM, UNet1D, CNN-based Autoencoder[31]). They were tested across the aforementioned five downstream tasks.

**Cardiovascular disease diagnosis (wearable ECG, PPG, 12-lead ECG).** The performance of CSFM was evaluated on three datasets representing distinct sensing modalities and acquisition channels: CinC17[22] (4 classes, multiclass classification), PTB-XL[21] (44 classes, multilabel classification) and SimBand[23] (4 classes, multiclass classification). Each dataset was split subject-wise (80% training, 10% validation, 10% testing). The best performance of CSFM compared with traditional methods is as follows: on CinC17, CSFM achieved a macro-F1 of 0.677 (95% confidence intervals (CI): 0.656, 0.699) versus 0.634 (95% CI: 0.558, 0.710); on PTB-XL, it obtained 0.357 (95% CI: 0.338, 0.377) versus 0.328 (95% CI: 0.296, 0.361); and on SimBand, it reached 0.398 (95% CI: 0.279, 0.516) versus 0.357 (95% CI: 0.324, 0.391). These results, reported in terms of macro-F1, are shown in Fig. 2a. In most cases, CSFM series substantially outperforms conventional learning strategies. Notably, on SimBand, although the CSFM-Large model achieves the best mean macro-F1 performance, its performance appears slightly more variable compared with other CSFM series. Further metrics are reported in Supplementary Information section 3.2.

**Demographic information recognition (lead II ECG, PPG).** This was evaluated on VitalDB dataset using a subject-wise data split (80% training, 10% validation, 10% testing). Model performance was assessed by MAE for age and BMI regression (lower is better) and by AUC for gender classification (higher is better). We train the model for these three

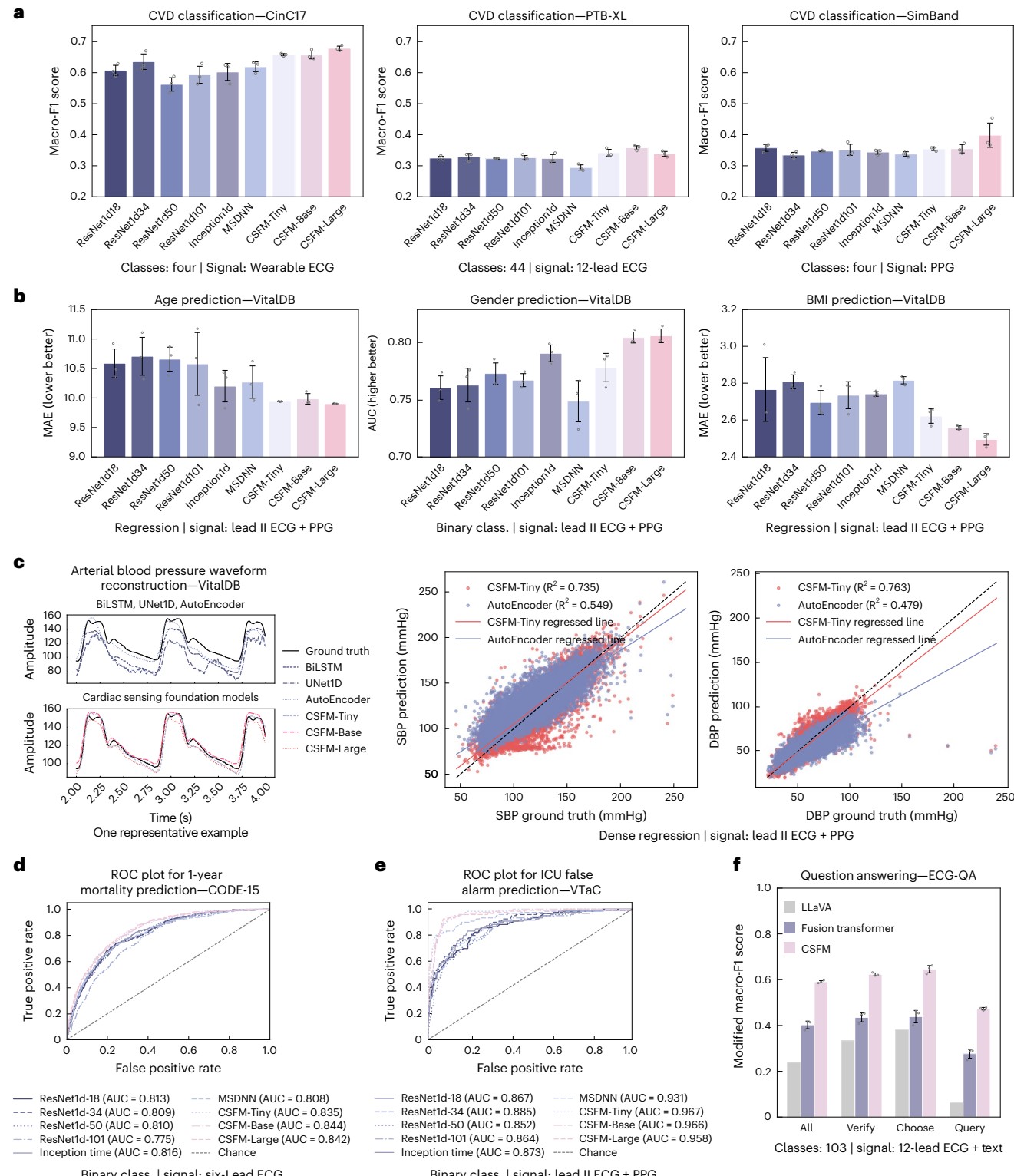

**Fig. 2 | Overall performance across different healthcare scenarios, validated on corresponding downstream datasets, separately. a**, Cardiovascular disease diagnosis across different datasets. The performance was measured by macro-F1 in terms of multilabel/class classification (class.). **b**, The demographic information recognition. Age and BMI prediction (univariate regression) were measured by MAE (lower is better), whereas gender prediction (binary class.) was measured by AUC (higher is better). **c**, The blood pressure waveform reconstruction based on lead II ECG and PPG as input. Left: one representative example. Right: the $R^2$ values of the derived SBP and DBP against the ground truths. **d**, The 1-year mortality prediction based on six-lead diagnostic ECGs.

The ROC curve of one run is presented. **e**, The ICU false alarm prediction based on signals (ECG and PPG) right before the alarm. The ROC curve of one run is presented. **f**, The ECG question answering with paired ECGs and questions. Question answering was formulated as a multichoice QA system in which, for each question template, the model selects the most appropriate answers from a set of candidate responses. All error-bar plots with **a**, **b** and **f** represent the mean ± s.d., computed over three independent runs with different random seeds ($n = 3$). Performance was measured using the macro-F1 score, computed over only the valid answers, as a modified macro-F1 score.

tasks separately. CSFM consistently outperforms models trained from scratch across all metrics, as presented in Fig. 2b.

**Vital sign measurement (lead II ECG, PPG).** We utilized VitalDB to evaluate blood pressure measurement performance under calibration-based settings, as standardized in the original paper[32]. We treat measurement predictions as a sequence-to-sequence regression problem. Specifically, CSFM first converts continuous ECG and/or PPG signals into continuous arterial blood pressure (ABP) signals and subsequently extracts the systolic (maximum, SBP) and diastolic (minimum, DBP) blood pressure values for comparison against ground truth. CSFM's performance was assessed in both stages, by comparing the waveform reconstruction quality as well as the numeric SBP and DBP values, as illustrated in Fig. 2c with $R^2$ value calculated. Further details including MAE can be found in Fig. 3.

**Clinical outcome prediction (ECG, PPG).** In this scenario, we evaluated the predictive performance of CSFM by forecasting the likelihood of adverse events. Specifically, we performed analysis in two distinct settings: first, for 1-year mortality prediction using six-lead ECGs (I, II, III, aVL, aVR and aVF as shown in Fig. 2d; results of other lead settings are in the next section), and second, in acute ICU environments where it was used to identify false ICU alarms on the basis of preceding ECG and PPG signals[25,33] (as shown in Fig. 2e). The former utilized the VTaC dataset[25], whereas the latter used the CODE-15 dataset[26], a publicly available subset of CODE-FULL. We split CODE-15 subject-wise (80% training, 10% validation, 10% testing) and VTaC on the basis of its official split and conducted binary classification for the outcome prediction on each dataset, respectively. It should be noted that CODE-15 is a public small version of CODE-Full[26], and in experimental settings, we ensured that no training subjects in CODE-Full were available in the validation or testing subset of CODE-15. These evaluations demonstrate the versatility of our approach across both immediate critical care and long-term risk stratification tasks. The receiver operating characteristic (ROC) curves, which illustrate the accuracy of our predictions, are presented in Fig. 2d,e. Additional results from three random seeds are available in Fig. 3a, showing that AUC for CSFM series reaches up to 0.844 for 1-year mortality prediction, compared with 0.816 for conventional methods trained from scratch. In addition, the false alarm prediction on VTaC achieves superior performance relative to traditional approaches, with best CSFM reaching AUC of 0.967 against 0.931 for traditional methods.

**ECG question answering (Text, ECG).** We leveraged the recently released ECG question answering benchmark, ECG-QA (PTB-XL version[27]), to test the model performance of answering specific ECG screening questions. Specifically, we assessed the model across three groups of tasks: single-verify, single-choose and single-query, each designed to probe different aspects of ECG interpretation. The expected answers from these questions are from a set of candidate templates, leading this question answering task to a multilabel classification task. Without a loss of generality, we selected CSFM-Tiny for comparison. The results are presented in Fig. 2f. This was compared with that of the Fusion Transformer model introduced in Oh et al.[27], as well as with LLaVA (llama3-llava-next-8b version), a large language model capable of image-text querying, which serves as the baseline. Further details are provided in Supplementary Information section 2.3. We reported macro-F1, which was computed per question by considering only the valid answer candidates for that question[27]. CSFM benefits from the pretraining, as demonstrated by its superior performance against the Fusion Transformer structure. The baseline performance of LLaVA was unsurprisingly limited, probably owing to its lack of domain-specific knowledge in ECG interpretation. More analysis results of ECG-QA are available in the next section.

Over the five scenarios examined, in certain cases, CSFM-Large, where applicable, occasionally exhibits slightly inferior performance compared with CSFM-Base. This observation suggests that its current dataset may be relatively insufficient to fully leverage the capacity of a larger model, in contrast to several existing large pretrained vision or language foundation models (for example, GPT4), which benefit from extensive pretraining on vast datasets. Future research may investigate the scaling laws of training foundation models in the cardiac biosignal domain to optimize the balance between model capacity and available data.

## CSFM generalizes across different ECG leads and ECG/PPG modalities

Ideally, to handle the varied lead and modality settings across care, the model should learn distinctive representations regardless of the type of cardiac signal provided as input and without additional architecture modifications. To test this capability, we evaluated performance using various channel and modality configurations.

**Performance under varied lead configurations of ECGs.** We assessed CSFM's transferability across different ECG lead configurations. In many diagnostic settings, standard 12-lead ECGs may not be readily available or affordable[34], posing considerable challenges for reusing models pretrained on specific lead configurations. CSFM, however, is designed to be adaptable across varied settings and demonstrates superior performance compared with conventional training methods. As shown in Fig. 3, experiments on PTB-XL (cardiovascular disease diagnosis) and CODE-15 (1-year mortality) confirm that CSFM outperforms existing bespoke models across varied lead configurations, including 12-lead, 6-lead (I, II, III, aVL, aVR and aVF), 2-lead (II and V5) and single-lead (lead II) set-ups.

**Performance under varied ECG and PPG Settings.** In addition, the availability of ECG and PPG can vary substantially in cardiac sensing applications. We examined our model's performance across scenarios with only ECG, only PPG or a combination of both when available. This evaluation was conducted using VTaC-based false alarm prediction (as shown in Fig. 3a, right) and VitalDB-based blood pressure reconstruction (as in Fig. 3b). Additional results of VitalDB-based demographic information recognition are provided in Supplementary Information section 3.3. CSFM consistently demonstrated superior performance across these different signal modalities compared with conventional methods. Overall, incorporating multiple channels led to higher performance, whereas it is also noted that PPG-only results in Fig. 3a (right), though lower in absolute terms, still exceeded baselines, reflecting the intrinsic difficulty of this modality for several specific tasks and datasets.

**Transfer from 12-lead to fewer-lead settings.** We further evaluate the transferability of models pretrained on 12-lead ECGs to fewer-leads. Specifically, we selected both conventional deep models and CSFM pretrained on PTB-XL (using 12-lead) and subsequently fine-tuned them on PTB-XL subsets with 6, 2 and 1 leads. We assessed performance when fine-tuning with 100%, 50% and 10% of the full training set of PTB-XL. This was similar to the protocols proposed in one previous work[35]. The results are presented in Fig. 3d. For conventional models, transferring pretrained weights to different lead configurations is not trivial, because their architectures often include layers with input channel sizes fixed to the number of leads (for example, the first one-dimensional (1D) convolutional layer in ResNet1d series is designed for 12 channels). To address this issue, we randomly reinitialized the weights of these input-specific layers during transfer learning, while transferring the remaining layers. By contrast, CSFMs are channel-agnostic, enabling direct transfer learning without the need to reinitialize input-specific layers. As observed in Fig. 3d, CSFMs consistently outperform conventional approaches. Notably, even when fine-tuning with only 10% of the training data, CSFM achieves performance comparable to conventional models trained on 100% of the dataset.

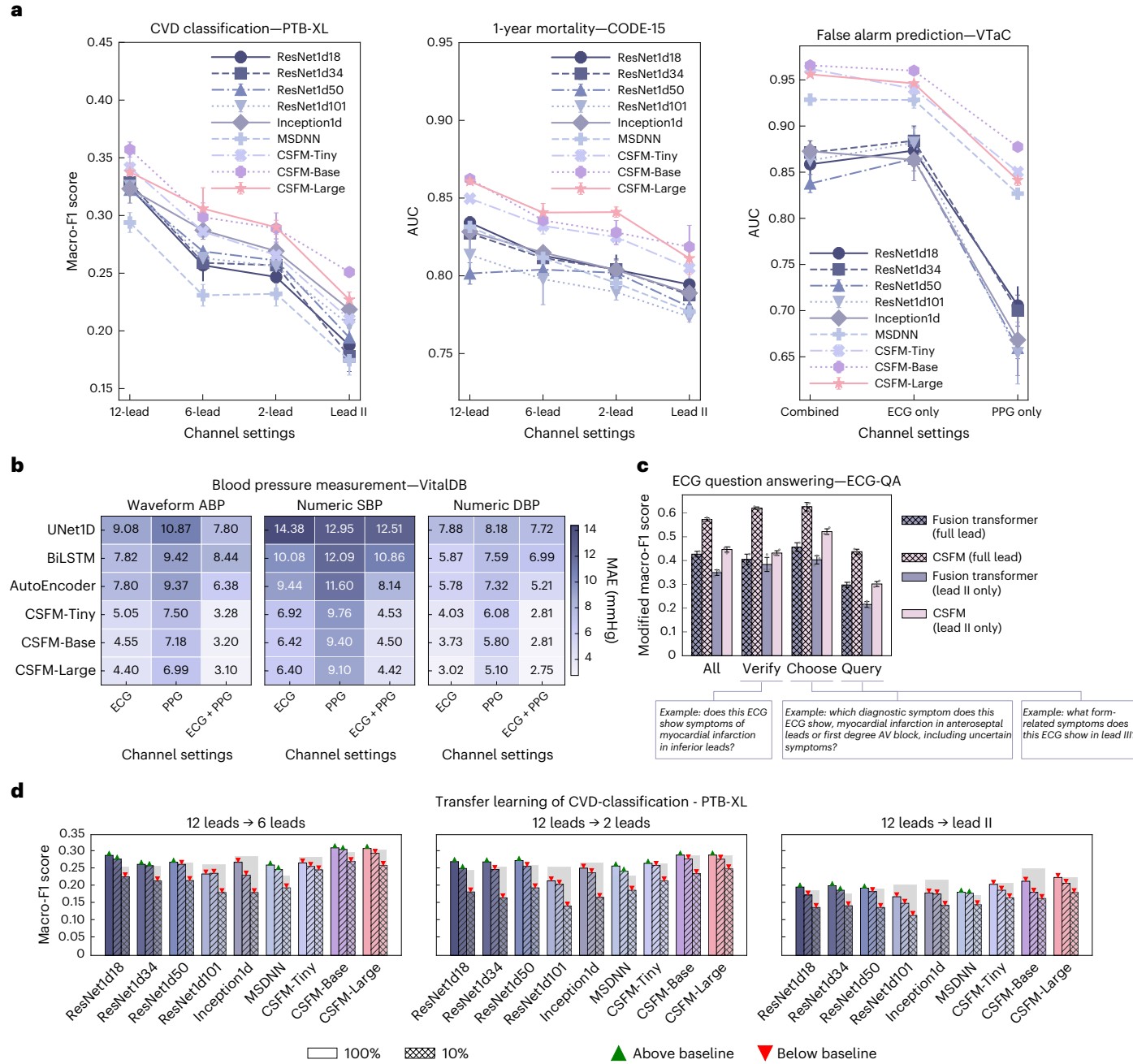

**Fig. 3 | Results of different ECG lead settings and ECG/PPG modalities. a**, The performance under different ECG channel settings (12-, 6-, 2-lead, lead II) for cardiovascular disease diagnosis (PTB-XL), 1-year mortality prediction (CODE-15) and ICU false alarm prediction (VTaC). **b**, The blood pressure prediction on VitalDB from continuous waveforms (ABP) and derived numeric values (SBP and DBP), evaluated by MAE (mmHg) across ECG, PPG or combined inputs. **c**, The ECG question answering (ECG-QA) for lead-related questions with only lead II as input, comparing Fusion Transformer and CSFM trained on lead II versus all 12 leads. Example questions are annotated. **d**, The transfer learning on PTB-XL with CSFM pretrained on 12-lead ECGs and fine-tuned on reduced-lead settings (6-, 2-, lead II) with 100%, 50% or 10% of training data. AV, atrioventricular. All error-bar plots within **a** and **c** represent the mean ± s.d., computed over three independent runs with different random seeds (*n* = 3). Bars denote test Macro-F1; shaded bars show direct 100% baselines; markers indicate performance gains (▲) or drops (▼).

**Lead-related ECG question answering with only lead II as input.**
We also examined whether CSFM can answer questions that typically depend on lead specifics (including keyword 'lead'), even when only a single lead (lead II) is provided as input. This represents a particularly challenging task, as most clinically relevant spatial patterns in ECG interpretation require multiple leads for accurate assessment, especially when questions involve features observable in other leads. The goal is to explore whether patterns typically distributed across multiple leads can, to some extent, be inferred from lead II alone. We compared the performance of CSFM and the Fusion Transformer under two input settings: full 12-lead ECG and lead II only. Notably, CSFM with only lead II input achieved performance comparable to that of the Fusion Transformer using the full 12-lead input. This suggests that CSFM's pretraining enables it to capture and uncover global information that generalizes beyond the visible lead.

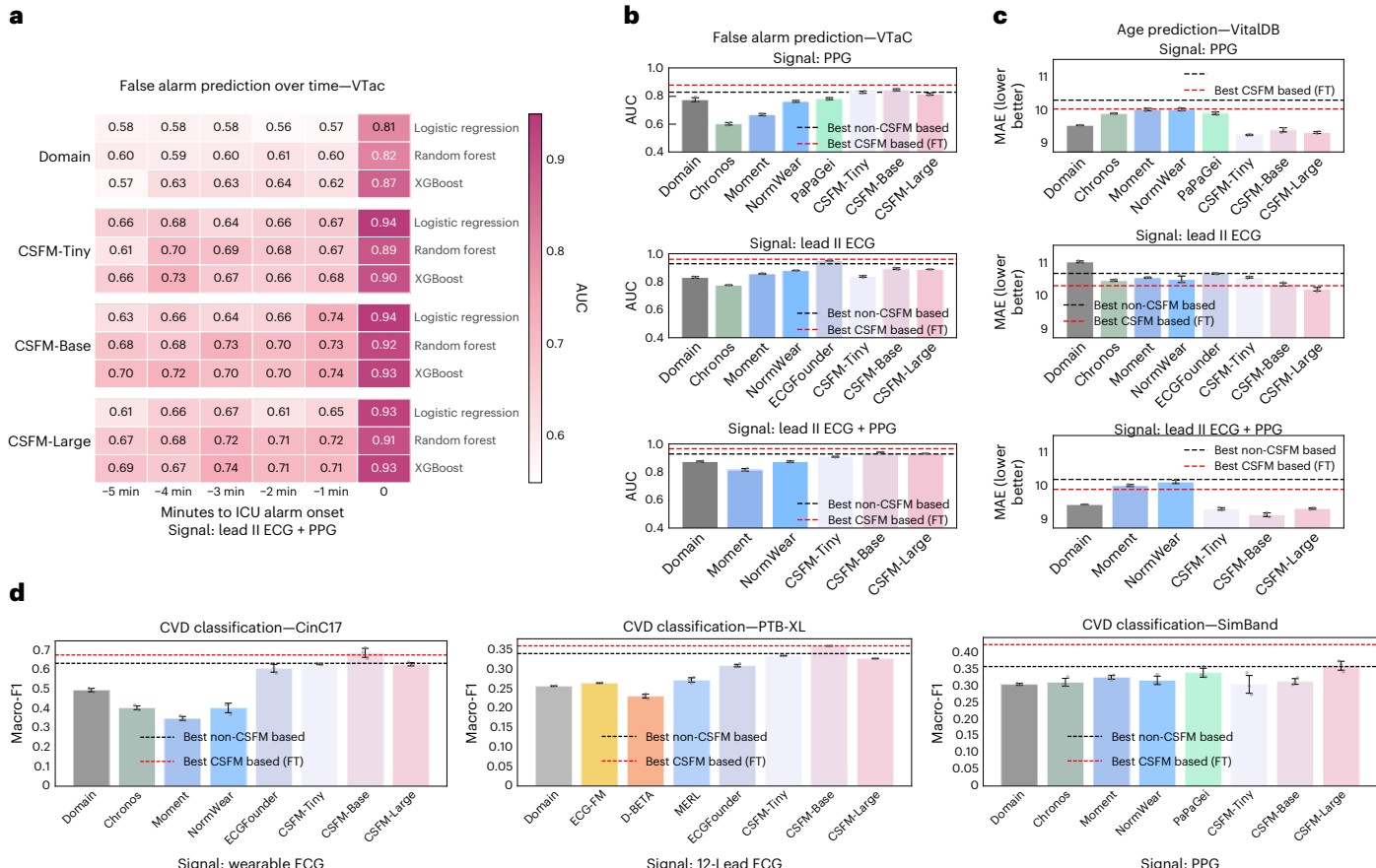

**Fig. 4 | Comparison of domain features, open-source foundation models and CSFM-derived features across multiple cardiac sensing tasks. a**, The predictive performance for ICU false alarm prediction on VTaC, evaluated using signals collected 0, 1, 2, 3, 4 and 5 min before alarm onset. Each feature set was assessed using logistic regression, random forest and XGBoost classifiers. **b**, A comparison of model performance on VTaC under different input modalities (PPG only, ECG only and combined ECG + PPG) between domain features, foundation-model-derived features and CSFM-derived features. **c**, The age prediction on VitalDB using PPG, lead II ECG and combined ECG + PPG signals, comparing domain features, foundation-model-derived features and CSFM-derived features. **d**, The cardiovascular disease classification across CinC17 (wearable ECG), PTB-XL (12-lead ECG) and SimBand (PPG) datasets. CVD, cardiovascular disease. All error-bar plots with **b**–**d** are represented as mean ± s.d., computed over three independent runs with different random seeds (*n* = 3). Across panels **b**–**d**, all feature sets were evaluated using XGBoost, and reference lines indicate the best-performing CSFM-based and non-CSFM-based methods, obtained either via fine-tuning or direct training, as reported in Figs. 2 and 3.

## CSFM acts as an effective feature extractor

The analysis of biomedical signals, particularly ECG and PPG, has advanced considerably over the years. The manual extraction of commonly used domain-specific features is still a popular approach. However, achieving robustness and generalizability across diverse data collection protocols and devices remains a critical challenge. Limited efforts have been made to develop a unified toolbox for extracting useful features from such heterogeneous settings, whether through traditional manual feature engineering or deep neural networks. We demonstrate that the representations derived by CSFM serve as effective embeddings, accommodating these variations and enhancing performance across settings.

**Comparison with manually engineered features.** First, we assessed the CSFM embeddings by comparing against domain-knowledge-driven bespoke features. To achieve this, we leveraged established biomedical-signal-processing toolboxes to extract relevant features from each modality separately. Specifically, we used NeuroKit2 (https://neuropsychology.github.io/NeuroKit/) and pyPPG (https://pyppg.readthedocs.io/) to extract features from ECG and PPG signals, respectively. Further details are available in Supplementary Information sections 2.1 and 3.4.

**Predictive performance over time horizons.** In addition, we benchmarked predictive performance for recognizing ICU false alarms over various time horizons. Specifically, we extracted 10-s recordings at multiple intervals (1, 2, 3, 4 and 5 min before alarms and immediately prior) and applied three classifiers (logistic regression, random forest and XGBoost) using both domain-specific features and CSFM embeddings. The results are displayed in Fig. 4a, indicating that CSFM embeddings consistently outperform domain-specific features. These suggest that CSFM can support the real-time recognition of early deterioration events, enabling timely clinical intervention.

**Comparison with features extracted from state-of-the-art time series foundation models.** Furthermore, we compared CSFM embeddings against those extracted from general time series models and from dedicated ECG/PPG foundation models. These are designed for time series or physiological signals, including Chronos[36], Moment[37], NormWear[38], PaPaGei[39], ECG-FM[40], ECGFounder[14], D-BETA[41] and MERL[42]. Each model differs in modality compatibility and signal dimensionality. Brief descriptions and implementation details for these models are summarized in Supplementary Information section 2.2. Their performance was assessed across multiple datasets and tasks, including PTB-XL, SimBand, CinC17 and VitalDB, evaluated using an XGBoost

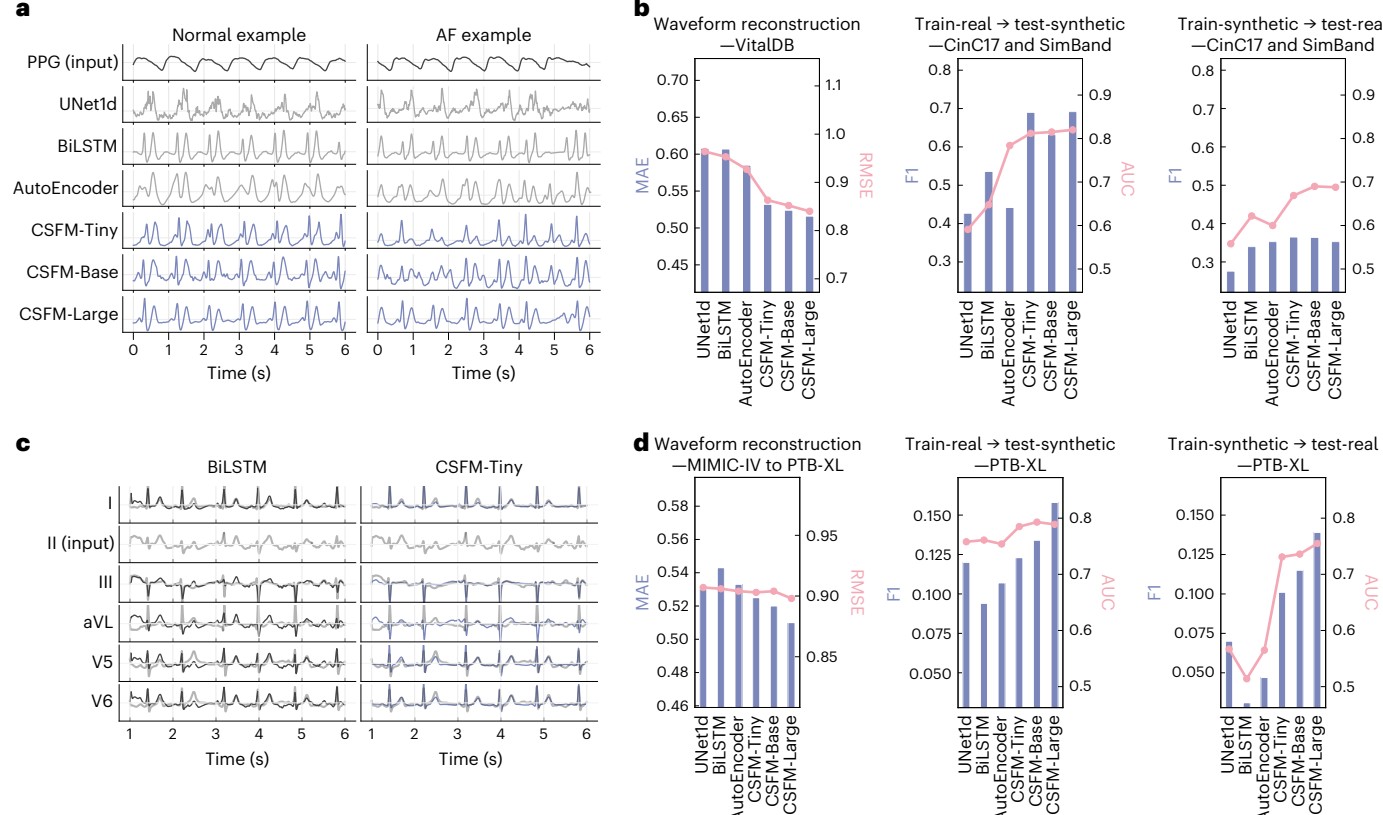

**Fig. 5 | Cross-modality reconstruction and augmentation results. a**, The representative examples of PPG-to-ECG reconstruction for both normal rhythm and AF. **b**, The quantitative evaluation of PPG-to-ECG reconstruction. Models were trained on VitalDB and applied to SimBand, with transfer evaluation on CinC17 (normal versus AF). Results are reported as MAE (bars) and root MSE (RMSE; lines) for waveform reconstruction and F1 (bars) and AUC (lines) for classification. **c**, The representative examples of single-lead to 12-lead ECG reconstruction, shown for cases of lateral wall ischaemia and non-specific ST/T changes. Predictions from BiLSTM (left) and CSFM-Tiny (right) are compared with ground truth (grey). Key lateral leads (I, aVL, V5, V6) are displayed. **d**, The quantitative evaluation of single-lead to 12-lead ECG reconstruction. Models were trained on MIMIC-IV (lead II input) and tested on PTB-XL under train-real/test-synthetic and train-synthetic/test-real conditions. Results are reported as MAE (bars) and RMSE (lines) for reconstruction and F1 (bars) and AUC (lines) for classification performance.

model. It should be noted that PaPaGei was pretrained on VitalDB, which may introduce potential data leakage when evaluated on the same dataset. As shown in Fig. 4b–d, modality-specific foundation models generally surpass general-purpose time series models, and we observed particularly strong performance from ECGFounder[14] on the VTaC lead II setting, probably owing to its pretraining on more than 10 million ECG recordings. Additional results for VitalDB-based BMI and gender prediction are provided in Supplementary Information section 3.3.

It is also noteworthy that, in several cases, CSFM embeddings, when used in conjunction with an XGBoost classifier, yield performance comparable to that of fully fine-tuned CSFM and conventional models trained from scratch (as reported in Figs. 2 and 3). This demonstrates the viability of directly using CSFM as a generic feature extractor for cardiac biosignals.

### CSFM facilitates cross-modality reconstruction and augmentation

Owing to the limitations of advanced sensing solutions, especially in several resource-limited scenarios, for example, low- and middle-income countries, collecting standard 12-Lead ECG is often challenging. This motivates our investigation into two specific applications to evaluate the versatility of CSFM. Similar to the vital sign measurement, we added a dense regression module on top of the transformer module to generate dense outputs.

**From PPG to ECG.** We generated ECG waveforms from PPG with CSFM and evaluated model's performance on atrial fibrillation (AF) detection. Figure 5a presents an example, and Fig. 5b summarizes the results, including both waveform reconstruction on the held-out test set of VitalDB and the transfer performances between synthetic ECGs generated from the SimBand dataset and real ECGs from CinC17. Specifically, we trained CSFM on VitalDB and then applied the trained model to the original SimBand dataset (selecting only normal and AF cases) to generate synthetic lead II ECG waveforms. To comprehensively evaluate the quality of these generated ECG waveforms, we assessed the transfer performance between the synthetic SimBand-ECG and CinC17, reporting performance metrics in terms of F1 and AUC.

**From single-lead ECG to 12-lead ECG.** We reconstructed full 12-lead ECGs from single-lead data, as synthetic data. The reconstruction model was trained on MIMIC-IV (lead II ECG → 12-lead ECG) and subsequently applied to PTB-XL to produce synthetic ECG recordings. We assessed the quality of these reconstructions under both train-real/test-synthetic and train-synthetic/test-real settings, with reconstruction examples and performance metrics provided in Fig. 5c,d.

Across these two tasks, the reconstructed data generated by CSFM demonstrate superior performance compared with the original data. However, a noticeable gap between real and synthetic data persists, as evidenced by the discrepancies observed between cross validations. Further analysis is available in Supplementary Information section 3.7.

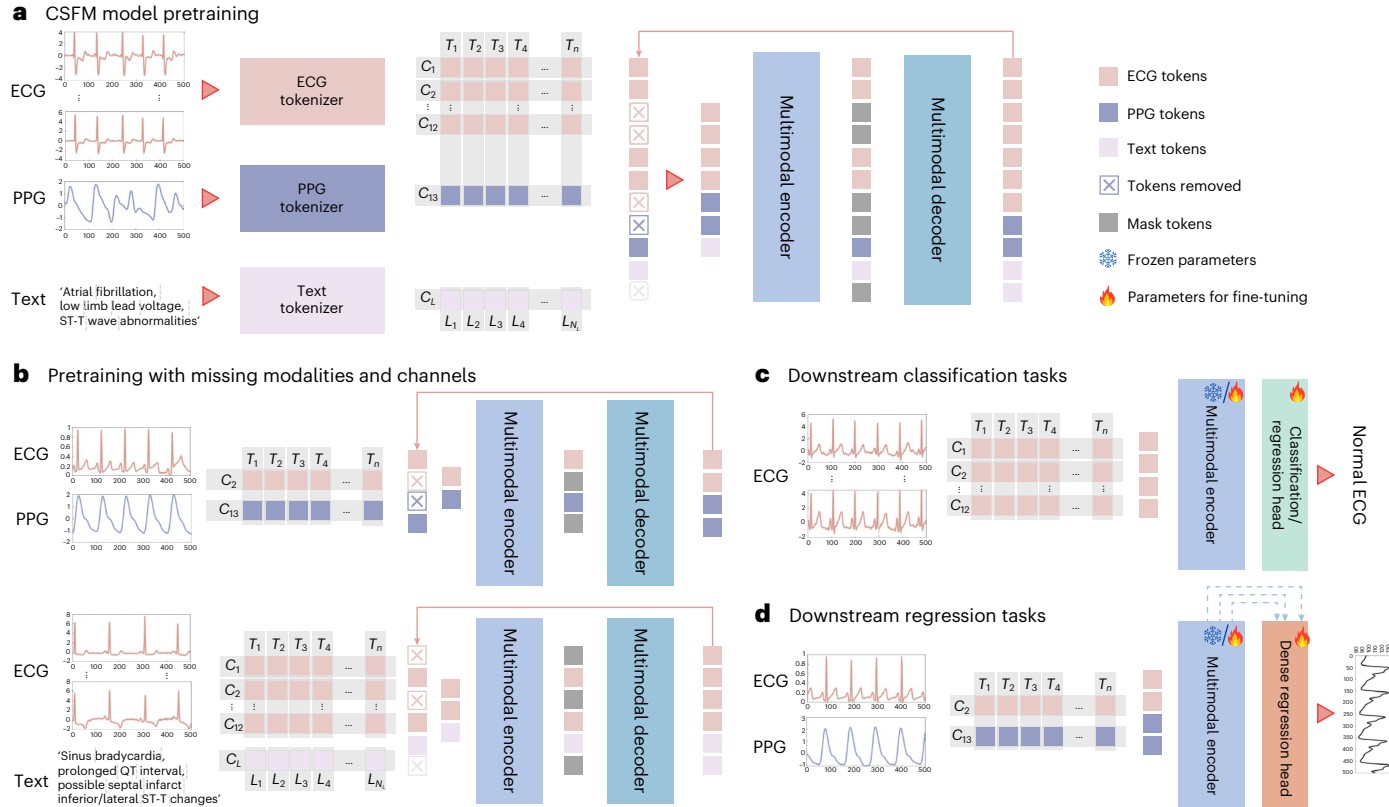

**Fig. 6 | Illustration of the whole framework. a**, The overall architecture of our framework, during pretraining. The input signals are first tokenized. The ECG and PPG signals are segmented into small non-overlapping patches followed by a shallow linear projector, whereas the text data are tokenized by WordPiece[44] and converted into learnable embedding vectors. They are subsequently added with learnable channel-wise and time/position-wise embeddings to encode modality-specific, and time- or position-specific information, separately. Following the standard practice of masked modelling, we randomly drop 75% of ECG tokens and 50% of text tokens and pass them through a Transformer-based encoder. The dropped tokens are subsequently replaced with mask tokens and fed into a decoder for reconstruction. **b**, The pretraining examples with missing modalities and channels: we flatten all tokens on the basis of the available signals and modalities, and perform reconstruction on the basis of available modalities. Subsequently, the loss was calculated on masked-out tokens. **c**, Downstream examples for classification or univariate regression: a shallow fully connected layer is appended to the encoder output to perform these tasks. **d**, Downstream examples for dense regression: a convolution-based dense regression module is applied on top of the encoder to progressively reconstruct the dense waveform from intermediate Transformer features. Blue arrows indicate feature aggregation across Transformer blocks.

Future work could explore conditional generative or diffusion models[43] to further enhance the plausibility and fidelity of generated signals.

## Discussion and conclusion

Our results demonstrate that CSFM robustly generalizes across a wide range of clinical scenarios, devices and input configurations. Its generalizability across tasks, versatility across signals and modalities, generative capability spanning resource-limited to resource-intensive settings and efficacy of embedding extraction collectively help bridge the current fragmentation of digital health systems, where models are often confined by modality-specific silos and narrow task optimization. Extensive evaluations across multiple datasets confirm that our model consistently outperforms conventional deep learning models and bespoke feature-based methods. Notably, the model-derived embeddings not only enhance diagnostic accuracy and predictive performance but also exhibit exceptional transferability across various lead configurations and sensing modalities.

Despite these promising results, our work has several limitations. First, the interpretability of deep models remains a challenge. Although CSFM captures intricate dependencies in cardiac biosignals, the 'black box' nature of its internal representations can limit clinical trust and adoption. Second, although we utilized the embeddings from existing language models, further integration with large language models with state-of-the-art methods, such as instruction tuning, may offer further improvements in interpretability and reasoning. Moreover, expanding the training to include larger and more diverse datasets would further enhance the robustness in certain settings, as well as uncover potential biases. Finally, the computational cost associated with training and deploying large-scale transformer architectures is non-trivial, potentially limiting accessibility in resource-constrained settings. Future work should enhance interpretability, reasoning and efficiency.

In conclusion, the CSFM represents an advancement in the analysis of heterogeneous cardiac biosignals. By leveraging advanced transformer architectures and a generative pretraining strategy on large-scale, diverse datasets (Fig. 6), our model learns robust, generalized representations that enhance diagnostic accuracy, predictive performance and transferability across varied sensor configurations and clinical scenarios.

Our comprehensive evaluations demonstrate that CSFM consistently outperforms traditional, modality-specific methods, offering a scalable solution adaptable to both resource-rich and resource-constrained settings. Although several challenges such as interpretability and computational cost remain, our findings underscore the potential of CSFM to transform cardiac monitoring and risk stratification. Overall, this work lays the groundwork for versatile cardiac monitoring tools poised to improve patient care and outcomes in cardiovascular medicine.

## Methods

### Heterogeneous input unification

To accommodate diverse modalities without relying on task-specific output structures, we use distinct tokenization strategies for signal and text inputs. These strategies transform the inputs to match the transformer input dimensionalities. Subsequently, the flattened tokens from both modalities are concatenated to form a unified input representation.

**Cardiac biosignals.** Let the cardiac biosignals from each individual $i$ be denoted as $\mathbf{x}_i \in \mathbb{R}^{C \times T}$, where $C$ represents the number of cardiac sensing channels, and $T$ denotes the total number of time steps. For a given health record $\mathbf{x}_i$, its channel set is defined as $C_i = \{c_{i_1}, c_{i_2}, ...\}$, where $C_i \subset C = \{c_1, c_2, ..., c_{|C|}\}$. Here, $C$ represents the full set of 12-lead ECG channels (that is, lead I, II, III, aVF, aVR, aVL, V1, V2, V3, V4, V5, V6) along with the additional PPG channel. In practice, we adopt lead II as the channel for single-lead wearable signals, following common practices in wearable ECG waveform analysis[7].

To transform the waveform into sequential tokens, we perform segmentation, linear projection and channel/temporal embedding addition. Specifically, each $\mathbf{x}_i$ is divided into non-overlapping segments of duration $w$ (that is, 0.1 s). This segmentation transforms $\mathbf{x}_i$ into patches $\mathbf{S}_i = \{\mathbf{s}_{i,j,k} | j = 1, 2, ..., C, k = 1, 2, ..., N_\mathbf{S}\}$, where $j, k$ index the channel and temporal order of the segments, respectively and $N_\mathbf{S} = \lfloor \frac{T}{w} \rfloor$ is the number of segments per signal. Subsequently, these patches are mapped into embeddings by a linear projection layer, resulting in $\mathbf{e}^\mathbf{S}_{i,j,k}$.

To incorporate information for both temporal and channel dimensions, we introduce positional embeddings for both dimensions. Specifically, we initialize a set of channel embeddings $\{\mathbf{e}^{\mathbf{S}_c}_j \in \mathbb{R}^d | j = 1, 2, ..., C\}$ and temporal embeddings $\{\mathbf{e}^{\mathbf{S}_T}_k \in \mathbb{R}^d | k = 1, 2, ..., \lfloor \frac{T}{w} \rfloor\}$. The final token embeddings are obtained by adding these positional embeddings to the projected embeddings, resulting in $\mathbf{E}^\mathbf{S}_i = \{\mathbf{e}_{i,j,k} + \mathbf{e}^{\mathbf{S}_c}_j + \mathbf{e}^{\mathbf{S}_T}_k | j = 1, 2, ..., C, k = 1, 2, ..., N_\mathbf{S}\}$.

**Text reports.** In addition to cardiac biosignals, the clinical workflow often incorporates textual data. Cardiologists may provide manual interpretations of these signals, which are recorded as clinical notes. By contrast, there are automated reports that may be generated by ECG machines. These textual interpretations provide supplementary insights into the cardiac signals and complement the overall analysis.

To process the textual data, each sentence is tokenized into a sequence of tokenized IDs, $\mathbf{L}_i = \{l_{i,k} | k = 1, 2, ..., N_\mathbf{L}\}$, on the basis of a pretrained language vocabulary using WordPiece[44]. Here, $N_\mathbf{L}$ denotes the number of text tokens (set as 64 in our study). The token IDs are then mapped to their corresponding embeddings using a learnable embedding layer. To align sentences of varied lengths, padding tokens are added to each sequence, leading to $\{\mathbf{e}^\mathbf{L}_{i,k} | k = 1, 2, ..., N_\mathbf{L}\}$. We further add additional text type embeddings $\mathbf{e}^{\mathbf{S}_\mathbf{L}}$ and learnable 1D positional embeddings $\{\mathbf{e}^\mathbf{L}_k\}$ to obtain the final input representations for the text as $\mathbf{E}^\mathbf{L}_i = \{\mathbf{e}^\mathbf{L}_{i,k} + \mathbf{e}^{\mathbf{S}_\mathbf{L}} + \mathbf{e}^\mathbf{L}_k | k = 1, 2, ..., N_\mathbf{L}\}$.

**Unification.** On the basis of the cardiac biosignals and, where available, the associated text reports, we introduce another class token $\mathbf{e}^{CLS}$ to extract the global feature by cross-attention across all available tokens. The final input to the model is represented as:

$$\mathbf{E}_i = \begin{cases} [\mathbf{e}^{CLS}; \mathbf{E}^\mathbf{S}_i], & \text{if text reports are unavailable,} \\ [\mathbf{e}^{CLS}; \mathbf{E}^\mathbf{S}_i; \mathbf{E}^\mathbf{L}_i], & \text{if both cardiac biosignals and text} \\ & \text{reports are available.} \end{cases} \quad (1)$$

In cases where text reports are unavailable, the model relies solely on $\mathbf{E}^\mathbf{S}_i$. When both sources are present, the embeddings are concatenated along the token dimension to form a unified input representation for our transformer model.

### Multimodal masked pretraining

Our masked pretraining framework, as illustrated in Fig. 6, is inspired by recent advancements in generative pretraining strategies, such as Masked Autoencoder[45]. For efficiency and simplicity, we randomly mask out 75% tokens from biosignals and 50% tokens from text. The superscription of $\mathcal{M}$ and $\overline{\mathcal{M}}$ denote the masked and unmasked tokens, respectively. The input to the model during pretraining is therefore $\mathbf{E}_i^{\overline{\mathcal{M}}}$. This high masking ratio greatly reduces redundant information, enabling the reconstruction objective to focus on intricate patterns in the remaining tokens. Furthermore, it reduces computational cost during pretraining by minimizing the number of tokens processed.

### Multimodal encoder

Our multimodal encoder is built upon the variant of Vision Transformer (ViT) architecture[12], which originally used a 'patchify' approach to divide the input image into fixed-size small patches and then transformed into a sequence of patch embeddings. Similarly, we process the unified input tokens (concatenating signals and text, when available) and pass them through a standard deep transformer. This encoder, is ultimately utilized as the CSFM for downstream validation, and the details of different CSFM scales are listed in Supplementary Table 7.

Throughout the multimodal encoder, these tokens are transformed into high-level multimodal representations, denoted as $\mathbf{z}^{CLS}$ (global representation), $\mathbf{z}^\mathbf{S}$ (signal-specific representation) and $\mathbf{z}^\mathbf{L}$ (text-specific representation). These embeddings capture the shared and modality-specific features necessary for downstream tasks.

### Multimodal decoder

To reconstruct the masked-out tokens, we combine the encoded visible tokens with a set of mask tokens, which act as placeholders for the decoder to reconstruct the original patches or word tokens at the masked positions[45]. As in the encoder, learnable channel-wise and temporal-wise positional embeddings are added to the tokens. The resulting embeddings are then processed by a lightweight multilayer perceptron and shallow Transformer blocks.

Following the original Masked AutoEncoder framework[45], we compute losses only for the masked tokens, thereby focusing the reconstruction objective on the unobserved parts of the input. This ensures efficient utilization of the masked modelling strategy while substantially reducing computational cost. The reconstruction process yields the restored signal patches, $\hat{\mathbf{S}}$, and text tokens, $\hat{\mathbf{L}}$.

The reconstruction loss is formulated as follows:

$$\mathcal{L}^\mathbf{S} = ||\mathbf{S}^\mathcal{M} - \hat{\mathbf{S}}^\mathcal{M}||_2, \quad (2)$$

$$\mathcal{L}^\mathbf{L} = -\log\left(p(\hat{\mathbf{L}}^{\overline{\mathcal{M}}} | \mathbf{E}^{\overline{\mathcal{M}}})\right), \quad (3)$$

where $\mathbf{S}^\mathcal{M}$ and $\hat{\mathbf{S}}^\mathcal{M}$ represent the ground truth and reconstructed signal patches at the masked positions, respectively. Similarly, $\hat{\mathbf{L}}^\mathcal{M}$ denotes the reconstructed masked text token indexes, conditioned on the visible embeddings $\mathbf{E}^{\overline{\mathcal{M}}}$.

### Downstream evaluation

A key capability of foundation models lies in their ability to derive generic representations from raw waveforms, which can be efficiently adapted to a variety of downstream tasks. To validate this capability, we evaluate the pretrained multimodal encoder across multiple tasks. Each task leverages different combinations of input data types, demonstrating the versatility of CSFMs.

**Evaluation tasks.** Task-wise, we classify the evaluation tasks into five distinct groups: demographic information recognition, cardiac disease classification, vital sign measurement, clinical outcome prediction and ECG question answering. These tasks broadly fall into two categories:

classification/regression tasks and dense sequence-to-sequence tasks. For consistency, we implement a standardized approach for all tasks within each group.

*Classification or univariate regression*. The majority of the tasks fall under the classification or univariate regression category. For these tasks, we extracted the representation of the global class token $e^{CLS}$ after it passed through all layers of the encoder. This representation is used as the input for an multilayer perceptron layer as an additional classifier, which may vary depending on the specific task requirements (such as class numbers). In terms of loss function, we applied asymmetric loss[46] for multilabel or multiclass classification tasks and mean squared error (MSE) for regression tasks.

*Dense sequence to sequence regression*. In the context of vital sign measurement, one specific task involves converting continuously recorded waveforms into blood pressure-related waveforms, such as ABP. Similarly, for cross-modality reconstruction tasks (for example, generating ECG from PPG or reconstructing a 12-lead ECG from a single lead II recording), a dense regression head is essential. Both types of task fall under the dense regression category. To accommodate this need, we adopted the dense prediction head proposed by Ranftl et al.[28], originally designed as a plug-and-play module for the ViT architecture. This module aggregates features from multiple transformer blocks to perform dense predictions, such as converting RGB images to depth maps. Built on this, the dense prediction head in CSFM was to map intermediate features from the input waveforms to the desired waveform outputs. The loss function for this task is defined using the MSE, ensuring accurate reconstruction of the target waveform.

**Utilization strategies of pretrained models.** We applied two strategies to evaluate the performance of our pretrained foundation models: either by freezing the pretrained model, as a deep feature extractor, or fine-tuning on the basis of the downstream annotations, tailored for the specific tasks.

*Feature extractor*. In this approach, the pretrained foundation model is frozen, and its representations are used as deep feature embeddings for downstream tasks. These features are passed to task-specific shallow predictors, such as logistic regression, XGBoost and random forest, which are trained exclusively on the annotations of the target tasks. Details are provided in Supplementary Information section 2.3.

*Fine-tuning*. In this context, the pretrained model is further fine-tuned on the downstream tasks by updating its parameters on the basis of task-specific annotations. We use layer-wise learning rate decay, where layers closer to the input use smaller learning rates compared with the deeper layers, with a decay rate of 0.75.

### Training and evaluation datasets

**Training datasets.** Pretraining was performed on an integrated dataset aggregated from multiple sources, including MIMIC-III-WDB[47], MIMIC-IV-ECG[18,19] and CODE-Full[19,47].

The first dataset, MIMIC-III-WDB, consists of waveform records collected from bedside patient monitors in ICUs at Beth Israel Deaconess Medical Center. These records were matched to the MIMIC-III clinical database, enabling the integration of waveform and clinical data. From this dataset, we extracted 10-s lead II ECG and PPG waveforms at 6-h intervals for each patient recording. To pair these waveforms with textual data, we utilized ECG notes available in the MIMIC-III clinical database, extracting ECG reports recorded within 1 week of each waveform segment. Notably, these notes are based on cardiologist-level interpretations of diagnostic 12-lead ECGs, rather than direct analyses of continuous ICU recordings. Consequently, this pairing is considered a 'weak match'.

The second dataset, MIMIC-IV-ECG, is a comprehensive collection of diagnostic ECGs associated with MIMIC-IV, a large-scale extension of MIMIC-III. We extracted both 12-lead diagnostic ECG waveforms and their paired summary reports, which were automatically generated by ECG machines. It is worth mentioning that there is an overlap in subjects between MIMIC-III and MIMIC-IV; however, both datasets are de-identified, making subject linkage currently impossible.

The third dataset, CODE-Full, is a private collection obtained from the Telehealth Network of Minas Gerais, Brazil[19,47]. This dataset comprises a substantial number of diagnostic ECG waveforms. For pretraining, we processed the data to create approximately 1.5 million segments by selecting one representative segment per subject.

The above data extraction results in around 1.7 million 10-s biomedical sensing signals collected from around 1.7 million individuals. The dataset was split into training (80%) and validation (20%) sets, ensuring no overlap between individuals in the two subsets. We applied basic filtering and *z*-score normalization of each biomedical signal. Please refer to the Supplementary Information for more details.

**Downstream evaluation datasets.** We evaluated the performance of the pretrained model across five distinct tasks using seven public datasets: VitalDB, PTB-XL, SimBand, CinC17, VTaC, CODE-15 and ECG-QA. For ECG-QA, VitalDB and VTaC, we followed their original training and testing splits, allocating 20% of the training split as the validation set. For PTB-XL, SimBand, CinC17 and CODE-15, we performed an 8:1:1 random split of the data on the basis of subject identities, ensuring no subject overlap across splits to mitigate potential data leakage issues. Notably, CODE-15 is a public subset of CODE-Full, and we, therefore, set its test split explicitly excluded from the pretraining stage of CODE-Full to ensure a fair evaluation.

### Experimental settings

**Implementation of pretraining.** All the experiments were performed on V100 GPUs with PyTorch. For pretraining, we applied AdamW (weight decay as $5 \times 10^{-2}$, betas as {0.9, 0.95}) with a learning rate of {$5 \times 10^{-4}$, $1 \times 10^{-3}$, $5 \times 10^{-4}$}, batch size as {256, 256, 128} for CSFM-Tiny, CSFM-Base and CSFM-Large, individually, to accommodate our computing capacity. In each iteration, the model performs uniform sampling from different datasets to form the training minibatch. We set the maximum number of training steps to 25,000,000, with early stopping based on validation performance every 2,500 steps. In addition, a cosine warmup decay schedule was applied during the initial 5% of training.

**Implementation of downstream tasks.** Below, we demonstrate the adoption of pretrained models on each specific task. All the experiments were conducted with three random seeds, and the mean performance across three seeds is reported. For conventional deep learning-based methods, we applied a batch size of 32 with a learning rate of $1 \times 10^{-3}$ for all settings. For our CSFM models, we applied $1 \times 10^{-4}$ for fine-tuning, with layer-wise decay set as 0.75. The optimizer was all set as AdamW.

*Cardiovascular disease diagnosis*. One particular usage of cardiac biosignals is to aid in the diagnosis of cardiovascular diseases. To demonstrate the performance of our foundation model on different diseases as well as cardiac sensing protocols, we conducted experiments on three datasets, including CinC17[22], SimBand[23] and PTB-XL[21]. CinC17 is a benchmark dataset designed for the detection of AF using single-lead wearable ECG signals. SimBand represents a real-world wearable dataset, with PPG segments annotated as AF or not. PTB-XL, by contrast, is a large diagnostic dataset containing 12-lead ECG recordings annotated with detailed diagnostic labels. These experiments allow us to evaluate the model's robustness across different modalities, lead configurations and disease categories.

*Demographics information recognition.* Biosignals, such as ECG and PPG, inherently encode subtle biological characteristics linked to demographic attributes such as age, gender and BMI. The ability to extract these attributes demonstrates the model's capability to capture and interpret underlying physiological variations while also aiding in mitigating potential biases and advancing personalized medicine. To evaluate the model's performance in recognizing these demographic variations, we conducted experiments to regress the latent representations individually to age, gender and BMI.

For this evaluation, we used the VitalDB dataset, following the preprocessing strategies and calibration-free data split protocol of PulseDB[32], ensuring no subject overlap between the training and testing splits.

*Vital sign measurement.* Several established numeric vital sign recordings, such as heart rate and $SpO_2$, can be derived directly from ECG and PPG signals using standard biomedical signal analysis toolkits. By contrast, there is growing interest in extracting blood pressure measurements from non-invasive continuous cardiac signals[32]. Leveraging a widely adopted pipeline, we extract both continuous ABP waveforms and numeric SBP and DBP values from PPG, ECG or their combinations[48]. This process involves first mapping the signals to ABP and then calculating the maximum and minimum values to derive SBP and DBP, respectively.

We validated this approach on the VitalDB dataset, using a calibration-based protocol, following the settings of PulseDB[32]. In the protocol, test segments were strictly excluded from the training process. Other segments from the same test subjects were included to simulate scenarios where partial calibration information is available.

*Clinical outcome prediction.* Predicting clinical outcomes has emerged as a critical application of cardiac biosignals, with growing research interest in leveraging these signals to assess associated risks[3,19]. Such advancements aim to facilitate the development of timely preventive strategies and optimize the allocation of healthcare resources. We leveraged CODE-15[47] for 1-year mortality prediction and VTaC[25] for false alarm prediction in the ICU.

Regarding the mortality with 1-year follow-up, the utilized dataset CODE-15 comprises 12-lead diagnostic ECG recordings, which provide a comprehensive representation of cardiac activity. For the false alarm prediction, we specifically extracted lead II ECG and PPG signals to evaluate their utility in predicting ICU false alarms.

*ECG question answering.* Question-answering systems have gained growing research interest in the medical domain owing to their potential to enhance clinical decision-making and interpretability. In this study, we leveraged the open-source ECG-QA dataset, which is built on the PTB-XL dataset, to evaluate our model's performance. We benchmarked the model on three tasks: single ECG verify, single ECG choose and single ECG query, which collectively assess the system's ability to interpret and reason with ECG data in various clinical contexts. For these tasks, the input comprises both the ECG signals and corresponding questions, with the goal of selecting the correct answers on the basis of the provided context and queries. This set-up evaluates the model's capacity to integrate waveform data with textual information to generate clinically relevant answers.

*Cross-modality reconstruction and augmentation.* We applied two settings. The first is PPG to ECG reconstruction. (1) The reconstruction was performed on VitalDB, and the waveform reconstruction performance was reported on the held-out test set of VitalDB. (2) Subsequently, we applied the adapted model to the original SimBand dataset to generate synthetic lead II ECG waveforms. To comprehensively test the quality of generated ECG waveforms, we conducted two experimental settings: train on synthetic ECG from SimBand (normal versus AF) and test on real ECG on CinC17 (normal versus AF), and vice versa, with an independent model ResNet1d-18 trained from scratch each time.

For single-lead ECG to 12-lead ECG augmentation, likewise, we leveraged MIMIC-IV (training set) to perform the reconstruction of lead II ECG to the full 12-lead ECG. Subsequently, we applied the trained model on PTB-XL to generate synthetic ECG recordings. The reconstruction performance is measured within the whole set of PTB-XL. On the basis of the synthetic recordings, we performed both train-real test-synthetic and train-synthetic test-real settings, on PTB-XL and its original split, to assess the quality of generated ECG signals. This cross-validation was also conducted by an independent ResNet1d-18, which was trained from scratch each time.

## Reporting summary

Further information on research design is available in the Nature Portfolio Reporting Summary linked to this article.

## Data availability

The pretraining dataset MIMIC-III-WDB[17] is available online (https://physionet.org/content/mimic3wdb-matched/1.0/) and their extensive clinical information (including subject-matched ECG reports) is available subject to the corresponding data usage agreement (https://physionet.org/content/mimiciii/). The MIMIC-IV-ECG dataset[18] is available online (https://physionet.org/content/mimic-iv-ecg/1.0/) as well. CODE-Full[26] is available for research under request (https://doi.org/10.17044/scilifelab.15169716) to co-authors, A.H.R. and A.L.P.R. Regarding downstream validation datasets, VitalDB[24] (https://github.com/pulselabteam/PulseDB) (preprocessed by PulseDB[32]), CODE-15[26] (https://paperswithcode.com/dataset/code-15), VTaC[25] (https://www.physionet.org/content/vtac/1.0/), PTB-XL[21] (https://physionet.org/content/ptb-xl/1.0.3/) and CinC17[22] (https://physionet.org/content/challenge-2017/1.0.0/) are all available online. Moreover, SimBand[23] (https://www.synapse.org/Synapse:syn23565056/wiki/608635) is available on the basis of access application, while ECG-QA[27] (https://github.com/Jwoo5/ecg-qa/tree/master/ecgqa/ptbxl, PTB-XL version) is available online with associated processing scripts. Source data are provided with this paper.

## Code availability

The pretrained model weights and the inference scripts are available via GitHub at https://github.com/guxiao0822/Cardiac-Sensing-FM and via Zenodo at https://doi.org/10.5281/zenodo.17803610 (ref. 49).

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

## Acknowledgements

D.A.C. was funded by an NIHR Research Professorship, a Royal Academy of Engineering Research Chair and the InnoHK Hong Kong Centre for Cerebro-cardiovascular Engineering (COCHE) and was supported by the National Institute for Health Research (NIHR) Oxford Biomedical Research Centre (BRC) and the Pandemic Sciences Institute at the University of Oxford. A.L.P.R. is supported in part by National Council for Scientific and Technological Development (CNPq), Minas Gerais State Foundation for Research Support (FAPEMIG), Innovation Center on Artificial Intelligence for Health (CIIA-S) and Institute for Health Assessment and Translation for Chronic and Neglected Diseases of High Relevance (IATS-CARE).

## Author contributions

D.A.C. conceived and supervised the project and revised the manuscript. X.G. conceived and designed the study, curated data, conducted experiments and data analysis and drafted the manuscript. W.T. conducted experiments and revised the manuscript. J.H. performed the data analysis and revised the manuscript. Z.L. curated data, conducted experiments and revised the manuscript. V.S., F.L., S.N.G., A.H.R., P.S., K.B., L.C. and A.L.P.R. significantly contributed to methodology design, result interpretation and manuscript revision and finalization.

## Competing interests

A.H.R. holds indirect equity shares at Einthoven Technologia LTDA. K.B. and P.S. are employees and shareholders of GSK plc. All the other authors declare no competing interests.

## Additional information

**Correspondence and requests for materials** should be addressed to Xiao Gu, Zhangdaihong Liu or David A. Clifton.

[1]Department of Engineering Science, University of Oxford, Oxford, UK. [2]Department of Mathematics, City University of Hong Kong, Hong Kong, China. [3]Hong Kong Center for Cerebro-Cardiovascular Health Engineering, Hong Kong, China. [4]Brain and Behaviour Lab, Imperial College London, London, UK. [5]School of Computer Science, University of Nottingham, Nottingham, UK. [6]Department of Information Technology, Uppsala University, Uppsala, Sweden. [7]GSK plc, London, UK. [8]Nuffield Department of Primary Care Health Sciences, University of Oxford, Oxford, UK. [9]Department of Internal Medicine, Faculdade de Medicina and Telehealth Center and Cardiology Service, Hospital das Clínicas, Universidade Federal de Minas Gerais, Belo Horizonte, Brazil. [10]Oxford Suzhou Centre for Advanced Research, University of Oxford, Suzhou, China. ✉e-mail: xiao.gu@eng.ox.ac.uk; zhang.liu@oscar.ox.ac.uk; david.clifton@eng.ox.ac.uk

# Reporting Summary

## Statistics

For all statistical analyses, confirm that the following items are present in the figure legend, table legend, main text, or Methods section.

| n/a | Confirmed | |
|---|---|---|
| ☐ | ☒ | The exact sample size (*n*) for each experimental group/condition, given as a discrete number and unit of measurement |
| ☐ | ☒ | A statement on whether measurements were taken from distinct samples or whether the same sample was measured repeatedly |
| ☒ | ☐ | The statistical test(s) used AND whether they are one- or two-sided<br>*Only common tests should be described solely by name; describe more complex techniques in the Methods section.* |
| ☒ | ☐ | A description of all covariates tested |
| ☒ | ☐ | A description of any assumptions or corrections, such as tests of normality and adjustment for multiple comparisons |
| ☒ | ☐ | A full description of the statistical parameters including central tendency (e.g. means) or other basic estimates (e.g. regression coefficient) AND variation (e.g. standard deviation) or associated estimates of uncertainty (e.g. confidence intervals) |
| ☒ | ☐ | For null hypothesis testing, the test statistic (e.g. *F*, *t*, *r*) with confidence intervals, effect sizes, degrees of freedom and *P* value noted<br>*Give P values as exact values whenever suitable.* |
| ☒ | ☐ | For Bayesian analysis, information on the choice of priors and Markov chain Monte Carlo settings |
| ☒ | ☐ | For hierarchical and complex designs, identification of the appropriate level for tests and full reporting of outcomes |
| ☐ | ☒ | Estimates of effect sizes (e.g. Cohen's *d*, Pearson's *r*), indicating how they were calculated |

*Our web collection on statistics for biologists contains articles on many of the points above.*

## Software and code

Policy information about availability of computer code

| Data collection | All datasets used in this study are publicly available from third-party repositories. Some datasets (e.g., CODE-Full and MIMIC-III-WDB) require credentialed access or data use agreements. Dataset links and access instructions are provided in the Data Availability section. |
|---|---|
| Data analysis | The pretrained model weights and inference examples has been submitted as supplementary materials, and will be made available https://github.com/guxiao0822/Cardiac-Sensing-FM upon acceptance. The were developed based on python 3.11, pytorch 2.4, neruokit2 0.2, scikit-learn 1.6. |

For manuscripts utilizing custom algorithms or software that are central to the research but not yet described in published literature, software must be made available to editors and reviewers. We strongly encourage code deposition in a community repository (e.g. GitHub). See the Nature Portfolio guidelines for submitting code & software for further information.

## Data

Policy information about availability of data

All manuscripts must include a data availability statement. This statement should provide the following information, where applicable:
- Accession codes, unique identifiers, or web links for publicly available datasets
- A description of any restrictions on data availability
- For clinical datasets or third party data, please ensure that the statement adheres to our policy

A summary of all training and test data used is clearly illustrated in the manuscript.

# Research involving human participants, their data, or biological material

Policy information about studies with <u>human participants or human data</u>. See also policy information about <u>sex, gender (identity/presentation), and sexual orientation</u> and <u>race, ethnicity and racism</u>.

| | |
|---|---|
| Reporting on sex and gender | This study used publicly available, de-identified datasets (e.g., MIMIC-III/IV, PTB-XL, CODE) containing physiological recordings and limited demographic information. Where provided, sex was included as a binary variable for analysis. No analyses were stratified by sex or gender beyond those available in the source datasets. |
| Reporting on race, ethnicity, or other socially relevant groupings | Race or ethnicity information was not included or available in the datasets analyzed. No grouping, categorization, or inference based on race or ethnicity was performed. |
| Population characteristics | The study analyzed de-identified cardiac biosignals and related metadata from adult patients and healthy volunteers collected across diverse clinical and geographic settings. Population details for each dataset (e.g., age, sex distribution, clinical conditions) are summarized in the Supplementary Information, Section S1. |
| Recruitment | No participants were directly recruited for this study. All data were obtained from established, de-identified open-access or licensed clinical databases with prior ethics approval at their host institutions. |
| Ethics oversight | The use of MIMIC-III, MIMIC-IV, PTB-XL, and VitalDB data complies with the respective institutional review boards and data-use agreements. For the privately held CODE dataset, approval was obtained under existing institutional protocols. No new data were collected for this study, and all analyses were conducted on anonymized data in accordance with institutional and journal ethical guidelines. |

Note that full information on the approval of the study protocol must also be provided in the manuscript.

# Field-specific reporting

Please select the one below that is the best fit for your research. If you are not sure, read the appropriate sections before making your selection.

☐ Life sciences    ☐ Behavioural & social sciences    ☐ Ecological, evolutionary & environmental sciences

For a reference copy of the document with all sections, see <u>nature.com/documents/nr-reporting-summary-flat.pdf</u>

# Life sciences study design

All studies must disclose on these points even when the disclosure is negative.

| | |
|---|---|
| Sample size | *Describe how sample size was determined, detailing any statistical methods used to predetermine sample size OR if no sample-size calculation was performed, describe how sample sizes were chosen and provide a rationale for why these sample sizes are sufficient.* |
| Data exclusions | *Describe any data exclusions. If no data were excluded from the analyses, state so OR if data were excluded, describe the exclusions and the rationale behind them, indicating whether exclusion criteria were pre-established.* |
| Replication | *Describe the measures taken to verify the reproducibility of the experimental findings. If all attempts at replication were successful, confirm this OR if there are any findings that were not replicated or cannot be reproduced, note this and describe why.* |
| Randomization | *Describe how samples/organisms/participants were allocated into experimental groups. If allocation was not random, describe how covariates were controlled OR if this is not relevant to your study, explain why.* |
| Blinding | *Describe whether the investigators were blinded to group allocation during data collection and/or analysis. If blinding was not possible, describe why OR explain why blinding was not relevant to your study.* |

# Behavioural & social sciences study design

All studies must disclose on these points even when the disclosure is negative.

| | |
|---|---|
| Study description | *Briefly describe the study type including whether data are quantitative, qualitative, or mixed-methods (e.g. qualitative cross-sectional, quantitative experimental, mixed-methods case study).* |
| Research sample | *State the research sample (e.g. Harvard university undergraduates, villagers in rural India) and provide relevant demographic information (e.g. age, sex) and indicate whether the sample is representative. Provide a rationale for the study sample chosen. For studies involving existing datasets, please describe the dataset and source.* |
| Sampling strategy | *Describe the sampling procedure (e.g. random, snowball, stratified, convenience). Describe the statistical methods that were used to predetermine sample size OR if no sample-size calculation was performed, describe how sample sizes were chosen and provide a rationale for why these sample sizes are sufficient. For qualitative data, please indicate whether data saturation was considered, and* |

| | |
|---|---|
| Data collection | Provide details about the data collection procedure, including the instruments or devices used to record the data (e.g. pen and paper, computer, eye tracker, video or audio equipment) whether anyone was present besides the participant(s) and the researcher, and whether the researcher was blind to experimental condition and/or the study hypothesis during data collection. |
| Timing | Indicate the start and stop dates of data collection. If there is a gap between collection periods, state the dates for each sample cohort. |
| Data exclusions | If no data were excluded from the analyses, state so OR if data were excluded, provide the exact number of exclusions and the rationale behind them, indicating whether exclusion criteria were pre-established. |
| Non-participation | State how many participants dropped out/declined participation and the reason(s) given OR provide response rate OR state that no participants dropped out/declined participation. |
| Randomization | If participants were not allocated into experimental groups, state so OR describe how participants were allocated to groups, and if allocation was not random, describe how covariates were controlled. |

# Ecological, evolutionary & environmental sciences study design

All studies must disclose on these points even when the disclosure is negative.

| | |
|---|---|
| Study description | Briefly describe the study. For quantitative data include treatment factors and interactions, design structure (e.g. factorial, nested, hierarchical), nature and number of experimental units and replicates. |
| Research sample | Describe the research sample (e.g. a group of tagged Passer domesticus, all Stenocereus thurberi within Organ Pipe Cactus National Monument), and provide a rationale for the sample choice. When relevant, describe the organism taxa, source, sex, age range and any manipulations. State what population the sample is meant to represent when applicable. For studies involving existing datasets, describe the data and its source. |
| Sampling strategy | Note the sampling procedure. Describe the statistical methods that were used to predetermine sample size OR if no sample-size calculation was performed, describe how sample sizes were chosen and provide a rationale for why these sample sizes are sufficient. |
| Data collection | Describe the data collection procedure, including who recorded the data and how. |
| Timing and spatial scale | Indicate the start and stop dates of data collection, noting the frequency and periodicity of sampling and providing a rationale for these choices. If there is a gap between collection periods, state the dates for each sample cohort. Specify the spatial scale from which the data are taken |
| Data exclusions | If no data were excluded from the analyses, state so OR if data were excluded, describe the exclusions and the rationale behind them, indicating whether exclusion criteria were pre-established. |
| Reproducibility | Describe the measures taken to verify the reproducibility of experimental findings. For each experiment, note whether any attempts to repeat the experiment failed OR state that all attempts to repeat the experiment were successful. |
| Randomization | Describe how samples/organisms/participants were allocated into groups. If allocation was not random, describe how covariates were controlled. If this is not relevant to your study, explain why. |
| Blinding | Describe the extent of blinding used during data acquisition and analysis. If blinding was not possible, describe why OR explain why blinding was not relevant to your study. |

Did the study involve field work?  ☐ Yes   ☒ No

# Reporting for specific materials, systems and methods

We require information from authors about some types of materials, experimental systems and methods used in many studies. Here, indicate whether each material, system or method listed is relevant to your study. If you are not sure if a list item applies to your research, read the appropriate section before selecting a response.

## Materials & experimental systems

| n/a | Involved in the study |
|-----|----------------------|
| ☒ ☐ | Antibodies |
| ☒ ☐ | Eukaryotic cell lines |
| ☒ ☐ | Palaeontology and archaeology |
| ☒ ☐ | Animals and other organisms |
| ☒ ☐ | Clinical data |
| ☒ ☐ | Dual use research of concern |
| ☒ ☐ | Plants |

## Methods

| n/a | Involved in the study |
|-----|----------------------|
| ☒ ☐ | ChIP-seq |
| ☒ ☐ | Flow cytometry |
| ☒ ☐ | MRI-based neuroimaging |

## Plants

| | |
|---|---|
| Seed stocks | *Report on the source of all seed stocks or other plant material used. If applicable, state the seed stock centre and catalogue number. If plant specimens were collected from the field, describe the collection location, date and sampling procedures.* |
| Novel plant genotypes | *Describe the methods by which all novel plant genotypes were produced. This includes those generated by transgenic approaches, gene editing, chemical/radiation-based mutagenesis and hybridization. For transgenic lines, describe the transformation method, the number of independent lines analyzed and the generation upon which experiments were performed. For gene-edited lines, describe the editor used, the endogenous sequence targeted for editing, the targeting guide RNA sequence (if applicable) and how the editor was applied.* |
| Authentication | *Describe any authentication procedures for each seed stock used or novel genotype generated. Describe any experiments used to assess the effect of a mutation and, where applicable, how potential secondary effects (e.g. second site T-DNA insertions, mosiacism, off-target gene editing) were examined.* |

