## [Peer Review File · Nature Machine Intelligence]

Cardiac Health Assessment Across Scenarios and Devices Using A Multi-Modal Foundation Model Pretrained on Heterogeneous Data from 1.7 Million Individuals

Corresponding Author: Dr Xiao Gu

Version 0:

Reviewer comments:

Reviewer #1

(Remarks to the Author)

The paper presents a multi-modal foundation model capable of handling multi-lead ECGs combined with PPG, as well as each modality independently, for downstream classification and regression tasks. The novelty lies in integrating ECG and PPG signals into a single foundation model and performing masked reconstruction of waveform and text tokens.

The strengths of this manuscript include the comprehensive set of methodological details that has been provided alongside the use of publicly available datasets and open sharing of analytic code. All of this allows for possible replication by other groups.

With that said, we have a number of major and minor concerns, as outlined below.

Major Concerns

1. Quality of data and interpretation of results

The paper claims robust performance when combining multi-lead ECGs with PPG, as well as when using each modality independently for downstream tasks. However, this claim is not supported by the evidence. The performance with PPG-only input is consistently poor, which is unsurprising given the very limited number of PPG samples (~11,000 individuals) used for training. Although the model outperforms PaPaGei (trained on a similarly small PPG cohort of ~13,000 individuals), this result cannot reasonably be taken as evidence of robustness for PPG. This work should be viewed not as evidence of robustness, but rather as highlighting the need for substantially larger and more diverse PPG datasets to develop a truly generalizable foundation model for PPG.

Moreover, the authors restrict their PPG-only evaluation to just three tasks: false alarm prediction for VTaC, blood pressure estimation, and CVD classification on SimBand, on all of which the model performs poorly. On the SimBand data, a macro-F1 of 0.30 on a 4-class task is only marginally better than chance (≈ 0.25), indicating limited discriminative performance. One suggestion is to perform the predictions on other tasks with PPG only, such as on VitalDB for age, gender, and BMI prediction. The authors do not specify the reason why they did not do so, and as it stands, the model can only be considered robust for ECGs, not for PPG.

Furthermore, to substantiate claims of a robust ECG foundation model, the authors should compare their approach against a few other existing ECG-only foundation models, rather than limiting the comparison to a single ECG-FM model. The discussion and conclusions could also use more detail regarding how this model performs compared to other foundational models. It is novel to try and combine features from various modalities and this should be commended, but there is certainly room for improvement for downstream results.

2. Lack of context

Although it is understood that this is not a clinical journal, the discussion still could do a lot more to elaborate on the clinical

significance of this model. That is, how could it be adapted by other groups, and what type of clinical impact could it have? Furthermore, there is little in the way of characterization of the datasets in terms of clinical and demographic factors that are important for real-world performance and furthermore could shed light on potential biases inherent in the embeddings. In addition, the Reporting Summary is devoid of pertinent information regarding demographic factors concerning the patient data that was used.

In addition, the paper certainly utilizes many different datasets, as summarized in Figure 1, which can be seen as a strength. However, it is difficult while reading the main text to keep track of the number of classes, type of dataset, and so on for all the different results, and so it is difficult to assess how successful the downstream results really are. This lack of clarity greatly hampers the potential impact of the results.

Minor Concerns

1. Classification metrics: Macro-F1 emphasizes correct classification (TP or TN) for each class individually while under-penalizing misclassifications (FP, FN). This makes it less reliable in imbalanced settings. However, metrics like Matthews Correlation coefficient (MCC), which account for all TP, TN, FP, and FN simultaneously, provide a more balanced evaluation. It is therefore suggested to report MCC scores alongside or instead of macro-F1.

2. Clinical Outcome Prediction: In the subsection of the results on clinical outcome prediction, the description of results lacks sufficient detail regarding the underlying dataset. Other subsections provide this context, and the same should be done here for consistency and clarity.

3. Feature Extraction and Methods: in the Supplemental Information, a comprehensive list should be provided of the domain features extracted from both PPG and ECG signals. If an exhaustive listing is not feasible, at least report the major features used. Additionally, the hyperparameters applied to the three classifiers should be explicitly listed, as this information is essential for reproducibility.

4. Language and Terminology Issues: The manuscript contains a few spelling and typographical errors (e.g., "biosignals" misspelled on page 14, "mean" incorrectly typed as "meas" on page 6 in the demographic information subsection). These should be corrected throughout. Furthermore, acronyms are not always defined prior to use (e.g., "MLP", "CVD" appeared without prior definition). In some cases, acronyms are redefined multiple times or used ambiguously; for instance, "MAE" is defined multiple times as "mean absolute error" and again later used to denote "masked autoencoder". This creates unnecessary confusion. Acronyms should be defined once, used consistently, and never overloaded with multiple meanings.

Notes regarding the figures and tables:

Figure 1

- The coloring of Figure 1 is confusing. It took a few views to understand the various shades of blue/gray.

Figure 2

- The color coding for panels (a) and (b) is too similar and may confuse readers. Distinct color schemes should be used. Likewise, in Figure 6, the colors representing different features are nearly indistinguishable, which makes interpretation difficult. Use clearly distinct colors or alternative visual encodings to improve readability.

Figure 3

- Figure labeling is inconsistent. While the first figure specifies classes, sample sizes, and signal types, subsequent figures omit such details. For consistency and clarity, we strongly recommend adopting a uniform style across all figures, and when possible, referencing the relevant tables or sections for this information in the captions.

- What is the class balance for Figure 3a? The Macro F1 Scores seem poor, especially for only four classes. Could the 44 classes be combined somehow? It may be too difficult of a task. Figure 3b needs to emphasize that VitalDB only uses a Lead II ECG when discussing the demographic recognition because otherwise the error/AUC is poor.

- Figure 3c is unclear whether the depicted waveform represents a single example trace or an average across waveform segments from the VitalDB dataset. Please clarify this in the caption.

Table 2

- Some context of what a good metric value would be for the dataset and classes would be helpful.

Table 3

- The metrics from the reconstruction from either PPG and a single lead are not great. Because AF can be distinguished with the human eye, could the authors speculate why the AUCs are not higher?

(Remarks on code availability)

Reviewer #2

(Remarks to the Author)

The paper presents a multi-modal foundation model capable of handling multi-lead ECGs combined with PPG, as well as each modality independently, for downstream classification and regression tasks. The novelty lies in integrating ECG and PPG signals into a single foundation model and performing masked reconstruction of waveform and text tokens.

The strengths of this manuscript include the comprehensive set of methodological details that has been provided alongside the use of publicly available datasets and open sharing of analytic code. All of this allows for possible replication by other groups.

With that said, we have a number of major and minor concerns, as outlined below.

Major Concerns

1. Quality of data and interpretation of results

The paper claims robust performance when combining multi-lead ECGs with PPG, as well as when using each modality independently for downstream tasks. However, this claim is not supported by the evidence. The performance with PPG-only input is consistently poor, which is unsurprising given the very limited number of PPG samples (~11,000 individuals) used for training. Although the model outperforms PaPaGei (trained on a similarly small PPG cohort of ~13,000 individuals), this result cannot reasonably be taken as evidence of robustness for PPG. This work should be viewed not as evidence of robustness, but rather as highlighting the need for substantially larger and more diverse PPG datasets to develop a truly generalizable foundation model for PPG.

Moreover, the authors restrict their PPG-only evaluation to just three tasks: false alarm prediction for VTaC, blood pressure estimation, and CVD classification on SimBand, on all of which the model performs poorly. On the SimBand data, a macro-F1 of 0.30 on a 4-class task is only marginally better than chance (≈ 0.25), indicating limited discriminative performance. One suggestion is to perform the predictions on other tasks with PPG only, such as on VitalDB for age, gender, and BMI prediction. The authors do not specify the reason why they did not do so, and as it stands, the model can only be considered robust for ECGs, not for PPG.

Furthermore, to substantiate claims of a robust ECG foundation model, the authors should compare their approach against a few other existing ECG-only foundation models, rather than limiting the comparison to a single ECG-FM model. The discussion and conclusions could also use more detail regarding how this model performs compared to other foundational models. It is novel to try and combine features from various modalities and this should be commended, but there is certainly room for improvement for downstream results.

2. Lack of context

Although it is understood that this is not a clinical journal, the discussion still could do a lot more to elaborate on the clinical significance of this model. That is, how could it be adapted by other groups, and what type of clinical impact could it have? Furthermore, there is little in the way of characterization of the datasets in terms of clinical and demographic factors that are important for real-world performance and furthermore could shed light on potential biases inherent in the embeddings. In addition, the Reporting Summary is devoid of pertinent information regarding demographic factors concerning the patient data that was used.

In addition, the paper certainly utilizes many different datasets, as summarized in Figure 1, which can be seen as a strength. However, it is difficult while reading the main text to keep track of the number of classes, type of dataset, and so on for all the different results, and so it is difficult to assess how successful the downstream results really are. This lack of clarity with regards to the results greatly hampers the potential impact of the results.

Minor Concerns

1. Classification metrics: Macro-F1 emphasizes correct classification (TP or TN) for each class individually while under-penalizing misclassifications (FP, FN). This makes it less reliable in imbalanced settings. However, metrics like Matthews Correlation coefficient (MCC), which account for all TP, TN, FP, and FN simultaneously, provide a more balanced evaluation. It is therefore suggested to report MCC scores alongside or instead of macro-F1.

2. Clinical Outcome Prediction: In the subsection of the results on clinical outcome prediction, the description of results lacks sufficient detail regarding the underlying dataset. Other subsections provide this context, and the same should be done here for consistency and clarity.

3. Feature Extraction and Methods: in the Supplemental Information, a comprehensive list should be provided of the domain features extracted from both PPG and ECG signals. If an exhaustive listing is not feasible, at least report the major features used. Additionally, the hyperparameters applied to the three classifiers should be explicitly listed, as this information is essential for reproducibility.

4. Language and Terminology Issues: The manuscript contains a few spelling and typographical errors (e.g., “biosignals” misspelled on page 14, “mean” incorrectly typed as “meas” on page 6 in the demographic information subsection). These should be corrected throughout. Furthermore, acronyms are not always defined prior to use (e.g., “MLP”, “CVD” appeared without prior definition). In some cases, acronyms are redefined multiple times or used ambiguously; for instance, “MAE” is defined multiple times as “mean absolute error” and again later used to denote “masked autoencoder”. This creates unnecessary confusion. Acronyms should be defined once, used consistently, and never overloaded with multiple meanings.

Notes regarding the figures and tables:

Figure 1

- The coloring of Figure 1 is confusing. It took a few views to understand the various shades of blue/gray.

Figure 2:

- The color coding for panels (a) and (b) is too similar and may confuse readers. Distinct color schemes should be used. Likewise, in Figure 6, the colors representing different features are nearly indistinguishable, which makes interpretation difficult. Use clearly distinct colors or alternative visual encodings to improve readability. Figure 3:

- Figure labeling is inconsistent. While the first figure specifies classes, sample sizes, and signal types, subsequent figures omit such details. For consistency and clarity, I strongly recommend adopting a uniform style across all figures, and when possible, referencing the relevant tables or sections for these information in the captions.
 - What is the class balance for Figure 3a? The Macro F1 Scores seem poor, especially for only four classes. Could the 44 classes be combined somehow? It may be too difficult of a task. Figure 3b needs to emphasize that VitalDB only uses a Lead II ECG when discussing demographic recognition because otherwise the error/AUC is poor.
 - Figure 3c is unclear whether the depicted waveform represents a single example trace or an average across waveform segments from the VitalDB dataset. Please clarify this in the caption.
- Table 2
- Some context of what a good metric value would be for the dataset and classes would be helpful.
- Table 3
- The metrics from the reconstruction from either PPG and a single lead are not great. Because AF can be distinguished with the human eye, could the authors speculate why the AUCs are not higher?

(Remarks on code availability)

Reviewer #3

(Remarks to the Author)

The paper presents a multi-modal foundation model capable of handling multi-lead ECGs combined with PPG, as well as each modality independently, for downstream classification and regression tasks. The novelty lies in integrating ECG and PPG signals into a single foundation model and performing masked reconstruction of waveform and text tokens.

The strengths of this manuscript include the comprehensive set of methodological details that has been provided alongside the use of publicly available datasets and open sharing of analytic code. All of this allows for possible replication by other groups.

With that said, we have a number of major and minor concerns, as outlined below.

Major Concerns

1. Quality of data and interpretation of results

The paper claims robust performance when combining multi-lead ECGs with PPG, as well as when using each modality independently for downstream tasks. However, this claim is not supported by the evidence. The performance with PPG-only input is consistently poor, which is unsurprising given the very limited number of PPG samples (~11,000 individuals) used for training. Although the model outperforms PaPaGei (trained on a similarly small PPG cohort of ~13,000 individuals), this result cannot reasonably be taken as evidence of robustness for PPG. This work should be viewed not as evidence of robustness, but rather as highlighting the need for substantially larger and more diverse PPG datasets to develop a truly generalizable foundation model for PPG.

Moreover, the authors restrict their PPG-only evaluation to just three tasks: false alarm prediction for VTaC, blood pressure estimation, and CVD classification on SimBand, on all of which the model performs poorly. On the SimBand data, a macro-F1 of 0.30 on a 4-class task is only marginally better than chance (≈ 0.25), indicating limited discriminative performance. One suggestion is to perform the predictions on other tasks with PPG only, such as on VitalDB for age, gender, and BMI prediction. The authors do not specify the reason why they did not do so, and as it stands, the model can only be considered robust for ECGs, not for PPG.

Furthermore, to substantiate claims of a robust ECG foundation model, the authors should compare their approach against a few other existing ECG-only foundation models, rather than limiting the comparison to a single ECG-FM model. The discussion and conclusions could also use more detail regarding how this model performs compared to other foundational models. It is novel to try and combine features from various modalities and this should be commended, but there is certainly room for improvement for downstream results.

2. Lack of context

Although it is understood that this is not a clinical journal, the discussion still could do a lot more to elaborate on the clinical significance of this model. That is, how could it be adapted by other groups, and what type of clinical impact could it have? Furthermore, there is little in the way of characterization of the datasets in terms of clinical and demographic factors that are important for real-world performance and furthermore could shed light on potential biases inherent in the embeddings. In addition, the Reporting Summary is devoid of pertinent information regarding demographic factors concerning the patient data that was used.

In addition, the paper certainly utilizes many different datasets, as summarized in Figure 1, which can be seen as a strength. However, it is difficult while reading the main text to keep track of the number of classes, type of dataset, and so on for all the

different results, and so it is difficult to assess how successful the downstream results really are. This lack of clarity greatly hampers the potential impact of the results.

Minor Concerns

1. Classification metrics: Macro-F1 emphasizes correct classification (TP or TN) for each class individually while under-penalizing misclassifications (FP, FN). This makes it less reliable in imbalanced settings. However, metrics like Matthews Correlation coefficient (MCC), which account for all TP, TN, FP, and FN simultaneously, provide a more balanced evaluation. It is therefore suggested to report MCC scores alongside or instead of macro-F1.
2. Clinical Outcome Prediction: In the subsection of the results on clinical outcome prediction, the description of results lacks sufficient detail regarding the underlying dataset. Other subsections provide this context, and the same should be done here for consistency and clarity.
3. Feature Extraction and Methods: in the Supplemental Information, a comprehensive list should be provided of the domain features extracted from both PPG and ECG signals. If an exhaustive listing is not feasible, at least report the major features used. Additionally, the hyperparameters applied to the three classifiers should be explicitly listed, as this information is essential for reproducibility.
4. Language and Terminology Issues: The manuscript contains a few spelling and typographical errors (e.g., "biosignals" misspelled on page 14, "mean" incorrectly typed as "meas" on page 6 in the demographic information subsection). These should be corrected throughout. Furthermore, acronyms are not always defined prior to use (e.g., "MLP", "CVD" appeared without prior definition). In some cases, acronyms are redefined multiple times or used ambiguously; for instance, "MAE" is defined multiple times as "mean absolute error" and again later used to denote "masked autoencoder". This creates unnecessary confusion. Acronyms should be defined once, used consistently, and never overloaded with multiple meanings.

Notes regarding the figures and tables:

Figure 1

- The coloring of Figure 1 is confusing. It took a few views to understand the various shades of blue/gray.

Figure 2

- The color coding for panels (a) and (b) is too similar and may confuse readers. Distinct color schemes should be used. Likewise, in Figure 6, the colors representing different features are nearly indistinguishable, which makes interpretation difficult. Use clearly distinct colors or alternative visual encodings to improve readability.

Figure 3

- Figure labeling is inconsistent. While the first figure specifies classes, sample sizes, and signal types, subsequent figures omit such details. For consistency and clarity, we strongly recommend adopting a uniform style across all figures, and when possible, referencing the relevant tables or sections for this information in the captions.
- What is the class balance for Figure 3a? The Macro F1 Scores seem poor, especially for only four classes. Could the 44 classes be combined somehow? It may be too difficult of a task. Figure 3b needs to emphasize that VitalDB only uses a Lead II ECG when discussing the demographic recognition because otherwise the error/AUC is poor.
- Figure 3c is unclear whether the depicted waveform represents a single example trace or an average across waveform segments from the VitalDB dataset. Please clarify this in the caption.

Table 2

- Some context of what a good metric value would be for the dataset and classes would be helpful.

Table 3

- The metrics from the reconstruction from either PPG and a single lead are not great. Because AF can be distinguished with the human eye, could the authors speculate why the AUCs are not higher?

(Remarks on code availability)

Reviewer #4

(Remarks to the Author)

The authors introduced a multi-modal foundation model (CSFM) for cardiac biosignal analysis, pre-trained on heterogeneous data using multiple large-scale datasets (MIMIC-III-WDB, MIMIC-IV-ECG, and CODE). The paper demonstrate the model's application across various cardiac sensing tasks, this research has large scale and innovative multi-modal application, which combines ECG, PPG, and textual reports to test across five different clinical scenarios with multiple datasets. It is also very relevant for real-world applicability, it addresses practical challenges in resource-constrained settings.

As the paper is well structured and with many technical innovations, the figures and tables are also clear with interpretable findings.

However, there are some limitations: the model is trained mostly on MIMIC datasets where data is from US based ICU

settings, there could be limited global population and unclear generalization to outpatient care, 2. The evaluation was done comparing all deep learning models, with transformer over other sequential models for this specific domain, would be beneficial to compare traditional sequential models and add some discussion of concurrent foundation model developments in healthcare 2. CSFM-Large sometimes performs worse than CSFM-Base, e.g., figure 3, maybe scaling up the parameters are not optimal 3. The model has large set of parameters, but limited analysis of inference time or computational requirements for clinical deployment.

Minor comments: There is significant performance differences between real and synthetic data (Table 3, suggesting quality gaps of synthetic data and real data. When evaluating the models, the paper sometimes used metrics which are not very clear: e.g., modified Macro-F1; There are also not much details about data preprocessing and limited details on signal preprocessing and quality control measures, e.g., "weak matching" between ECG signals and clinical notes in MIMIC-III-WDB could introduce noise in the multi-modal learning process. Some figures have overlapping text(fig 8)

(Remarks on code availability)

The code is organized well with clear structure and documentation of README with clear installation instructions and usage examples, the author also provided a tutorial notebook with step-by-step guidance and separated code part of model definition, data loading, and training. The only limitation is some parameters are hard coded and may limit the model flexibilities.

Version 1:

Reviewer comments:

Reviewer #1

(Remarks to the Author)

The authors should be commended for this very comprehensive response to our comments. We are of the view that all of our concerns have been addressed, and the manuscript is now of sufficient quality to warrant publication in Nature Machine Intelligence.

(Remarks on code availability)

Reviewer #2

(Remarks to the Author)

We appreciate the authors' comprehensive revisions in response to our comments. It is our assessment that the concerns raised have been satisfactorily addressed, and the manuscript is now suitable for publication in Nature Machine Intelligence.

(Remarks on code availability)

Reviewer #3

(Remarks to the Author)

The authors have addressed my concerns, and I have no further comments.

(Remarks on code availability)

Reviewer #4

(Remarks to the Author)

Thank you for addressing my previous comments. I've read the revised version of the manuscript and the feedback provided by the other reviewers. The authors have made substantial improvements in response to the earlier concerns, and the current version is well-written and suitable for publication.

(Remarks on code availability)

Response to review comments

NATMACHINTELL-A25073100-A

We thank the reviewers for their thorough reading of our manuscript and for providing constructive feedback to further improve our work. Extensive modifications have been made to the paper based on the comments received, and we have attached herewith a point-by-point explanation of all the changes made to the manuscript. The text highlighted in blue corresponds to the revised text or figure/table number in the paper.

In summary, the major changes made to the paper include:

- **Expanded baselines and comparisons** (C 1.1,C 1.2,C 4.2): Benchmarked against a broader range of ECG-, PPG-, and time-series foundation models, including both modality-specific and general architectures.
- **Extended PPG-only evaluations** (C 1.1): Expanded PPG-only experiments and added ablation studies.
- **Clinical relevance and context** (C 1.3,C 1.4,C 4.1,C 4.4): Clarified clinically related settings, applications, and limitations; incorporated dataset demographics and efficiency analysis relevant to deployment.
- **Clarification of metrics, datasets, and experimental settings** (C 1.3,C 1.5,C 1.6,C 1.7, C 1.12,C 4.3,C 4.5,C 4.6): Clarified evaluation metrics, dataset usage, and experimental protocols; reported additional metrics.
- **Extended analyses** (C 1.7,C 1.8,C 1.13,C 4.5,C 4.7): Conducted further analyses of domain features, PTB-XL results, PPG to ECG generation, and weak text–signal pairing.
- **Enhanced figures and presentation** (C 1.9,C 1.10,C 1.11): Standardized labelings, and improved overall figure accessibility and coherence.

All minor comments and typographical errors, as well as editorial compliances have been addressed and rectified in the revised manuscript.

In the following we address the comments point by point.

Editor

C E.1 — Please revise the title, as no punctuation is allowed.

Reply: Thanks, we have changed the title to "Sensing Cardiac Health Across Scenarios and Devices Using A Multi-Modal Foundation Model Pretrained on Heterogeneous Data from 1.7 Million Individuals", by removing the colon ":".

C E.2 — The abstract should start with 2 or 3 sentences describing the background and context of the work and should be around 220 words in length.

Reply: Thanks, we have updated the abstract to the following:

Abstract

Cardiovascular diseases remain a major contributor to the global burden of healthcare, highlighting the importance of accurate and scalable methods for cardiac monitoring. Cardiac biosignals, most notably electrocardiograms (ECG) and photoplethysmograms (PPG), are essential for diagnosing, preventing, and managing cardiovascular conditions across clinical and home settings. However, their acquisition varies substantially across scenarios and devices, while existing analytical models often rely on homogeneous datasets and static bespoke models, limiting their robustness and generalizability in diverse real-world contexts. In this study, we present a cardiac sensing foundation model (CSFM) that leverages transformer architectures and a generative masked pretraining strategy to learn unified representations from heterogeneous health records. CSFM is pretrained on a multimodal integration of data from various large-scale datasets, comprising cardiac signals from approximately 1.7 million individuals and their corresponding clinical or machine-generated text reports. The embeddings derived from CSFM act as effective, transferable features across diverse cardiac sensing scenarios, supporting seamless adaptation to varied input configurations and sensor modalities. Extensive evaluations across diagnostic tasks, demographic recognition, vital sign measurement, clinical outcome prediction, and ECG question answering demonstrate that CSFM consistently outperforms traditional one-modal-one-task approaches. Notably, CSFM maintains favorable performance across both 12-lead and single-lead ECGs, as well as in scenarios involving ECG only, PPG only, or a combination of both. This highlights its potential as a versatile and scalable foundation for comprehensive cardiac monitoring.

Word Count: 225

C E.3 — Please remove references to your results and methodology being 'novel' or 'new'.

Reply: Thank you, we have reviewed the whole paper, and removed such phrases as appropriate.

C E.4 — The length of the main article (without abstract or methods section) should be below 4000 words.

Reply: Thank you, the current word counts are 3998, which has been explicitly mentioned in the cover letter.

C E.5 — - The maximum number of display items is 6.

- Please remove cartoons and icons from figures 1 and 2.

Reply: Thank you, we have rearranged the presentations, so that there are 6 display items in total in the main paper. In addition, we have minimized the use of icons and removed all cartoon elements in the revised Figure 1 (corresponding to the original Figures 1 and 2).

C E.6 — Data availability and code availability

Reply: Thank you, we have clearly set sections related to data availability and code availability settings, as required. For the code, we provided the link of the codebase, which will be made open-source upon acceptance; the current version has been provided as part of the Supplementary Materials.

Reviewer 1,2,3

C 1.1 — Quality of data and interpretation of results 1/2

The paper claims robust performance when combining multi-lead ECGs with PPG, as well as when using each modality independently for downstream tasks. However, this claim is not supported by the evidence. The performance with PPG-only input is consistently poor, which is unsurprising given the very limited number of PPG samples (11,000 individuals) used for training. Although the model outperforms PaPaGei (trained on a similarly small PPG cohort of 13,000 individuals), this result cannot reasonably be taken as evidence of robustness for PPG. This work should be viewed not as evidence of robustness, but rather as highlighting the need for substantially larger and more diverse PPG datasets to develop a truly generalizable foundation model for PPG.

Moreover, the authors restrict their PPG-only evaluation to just three tasks: false alarm prediction for VTaC, blood pressure estimation, and CVD classification on SimBand, on all of which the model performs poorly. On the SimBand data, a macro-F1 of 0.30 on a 4-class task is only marginally better than chance (≈ 0.25), indicating limited discriminative performance. One suggestion is to perform the predictions on other tasks with PPG only, such as on VitalDB for age, gender, and BMI prediction. The authors do not specify the reason why they did not do so, and as it stands, the model can only be considered robust for ECGs, not for PPG.

Reply: We thank the reviewer for raising this important point.

PPG is, in many respects, a more “challenging” modality, if not equally so, compared with ECG, since its signals are inherently noisier, less standardized across devices. And we agree that the relatively modest PPG cohort size (around 270k samples, around 7k individuals) limits the achievable performance, compared to what may be achievable by the “gold standard” ECG. We anticipate that further improvements would be possible if more diverse PPG datasets were available for training.

To address this concern and better clarify our contributions, we have revised the manuscript in the following four aspects:

a. Expanded baseline comparisons.

We thank the reviewer for this valuable suggestion. We agree that broader evaluations provide a clearer understanding of our model’s capabilities and limitations in handling PPG data.

In addition to existing PPG-only comparisons, we have expanded the evaluation to include a wider range of models and downstream task settings, covering both fine-tuning and feature-extraction configurations.

Model-wise, we have added NormWear [1] (designed specifically for multivariate physiological signals) and general time-series foundation models for comparison, including Chronos [2] and Moment [3]. This is added in Figure 2b–d (Figure 4b–d in the main paper) and Figure 1 (Supplementary Figure S2 in the Supplementary Material).

Task-wise, we have also included comparisons for demographic information recognition under both fine-tuning and feature-extraction settings. This is added in Figure 1 (Supplementary Figure S2), with age-prediction results additionally presented in the main paper Figure 2c (Figure 4c).

Across all settings, our model achieves consistent improvements, underscoring that while some challenges arise from the PPG modality itself, our framework is able to extract more informative PPG representations than existing approaches.

Figure 1: **Additional results on VitalDB.** **a.** Age, gender, and BMI prediction performance under different input-channel configurations (Lead-II ECG-only, PPG-only, or combined), comparing fine-tuned CSFM models with directly trained non-CSFM deep models. **b.** Performance comparison based on domain features, foundation-model-derived features, and CSFM-derived features across age, gender, and BMI prediction tasks, based on XGBoost, and reference lines indicate the best-performing CSFM-based and non-CSFM-based methods, reported in subfigure **a**.

Figure 2: **Comparison of domain features, open-source foundation models, and CSFM-derived features across multiple cardiac sensing tasks.** **a.** Predictive performance for ICU false alarm prediction on VTaC, evaluated using signals collected 0, 1, 2, 3, 4, and 5 minutes prior to alarm onset. Each feature set was assessed using Logistic Regression, Random Forest, and XGBoost classifiers. **b.** Comparison of model performance on VTaC under different input modalities (PPG only, ECG only, and combined ECG+PPG) between domain features, foundation-model-derived features, and CSFM-derived features. **c.** Age prediction on VitalDB using PPG, Lead-II ECG, and combined ECG+PPG signals, comparing domain features, foundation-model-derived features, and CSFM-derived features. **d.** Cardiovascular disease classification across CinC17 (wearable ECG), PTB-XL (12-lead ECG), and SimBand (PPG) datasets. Across panels **b–d**, all feature sets were evaluated using XGBoost, and reference lines indicate the best-performing CSFM-based and non-CSFM-based methods, obtained either via fine-tuning or direct training, as reported in Figures 6 and 3.

Figure 3: **Results of different ECG lead settings and ECG/PPG modalities.** **a.** Performance under different ECG channel settings (12-, 6-, 2-lead, Lead II) for cardiovascular disease diagnosis (PTB-XL), 1-year mortality prediction (CODE-15), and ICU false alarm prediction (VTaC). **b.** Blood pressure prediction on VitalDB from continuous waveforms (arterial blood pressure, ABP) and derived numeric values [systolic (SBP), diastolic (DBP)], evaluated by mean absolute error (mmHg) across ECG, PPG, or combined inputs. **c.** ECG question answering (ECG-QA) for lead-related questions with only Lead II as input, comparing Fusion Transformer and CSFM trained on Lead II versus all 12 leads. Example questions are annotated. **d.** Transfer learning on PTB-XL with CSFM pretrained on 12-lead ECGs and fine-tuned on reduced-lead settings (6-, 2-, Lead II) with 100%, 50%, or 10% of training data. Bars denote test Macro-F1; shaded bars show direct 100% baselines; markers indicate performance gains (\blacktriangle) or drops (\blacktriangledown).

b. Heterogeneous multi-modal pretraining enables PPG.

Although the diversity of PPG data itself is limited, our heterogeneous pretraining across ECG, PPG, and text leverages large amounts of non-PPG modalities, which significantly enhances PPG representation learning. To examine this effect, we conducted additional experiments where PPG was pretrained (i) in isolation, (ii) with paired ECG, (iii) with paired ECG and text, and (iv) with all modalities. Results show that joint pretraining with ECG and text consistently improves downstream PPG performance compared to PPG-only training. This demonstrates that multimodal pretraining allows knowledge transfer from ECG to PPG, partially compensating for the limited scale and diversity of PPG data.

The results are shown in the Supplementary Material,

Supplementary S3.6 **“Non-PPG modalities facilitate PPG representation learning.** Furthermore, it is noteworthy that during pretraining only MIMIC-III-WDB includes PPG dataset, comprising approximately 270k PPG segments (7k individuals). We conducted experiments to validate whether additional modalities from ECG, associated clinical text, and even non-paired 12-lead ECGs/texts can facilitate the pretraining of a model that learns better representations of PPG. The results in Figure 13b show that models pretrained with a greater diversity of modalities achieve progressively better performance across five PPG-related tasks, demonstrating that richer multimodal supervision leads to more informative representations.”

c. Reframing the narrative.

Based on the expanded results, we have carefully revised the manuscript text to avoid overclaiming robustness in the PPG-only setting and to provide more detailed analysis of the challenges.

“Main Page 6 Overall, incorporating multiple channels led to higher performance, while it is also noted that PPG-only results in the right panel of Figure 3a, though lower in absolute terms, still exceeded baselines, reflecting the intrinsic difficulty of this modality for some specific tasks/datasets.”

We also emphasize that our findings highlight the need for larger and more diverse PPG datasets in order to build truly generalizable PPG foundation models, while positioning our results as evidence of the feasibility and promise of multimodal transfer.

Main Page 10 “Despite these promising results, ... Moreover, expanding training to include larger and more diverse datasets would further enhance robustness in certain settings. ...”

d. Clarification of performance limit.

The relatively modest performance on the SimBand dataset is likely constrained by the data itself, which is highly imbalanced and includes only 40 subjects, as now clarified in the Supplementary Materials.

In addition to the macro-F1 score, we now also report the Matthews Correlation Coefficient (MCC) in Table 8 (Table S5 in the Supplementary), in accordance with C1.6. The MCC results demonstrate that the superior performance of CSFM in the PPG-only setting; thus the performance might be limited by the dataset characteristics per se.

Figure 4: **Ablation study for different pretraining settings.** **a.** Comparison of our training strategies against other “straightforward” solutions to handling heterogeneous health records, (i) keeping the common channels only, *i.e.*, Lead-II ECG, (ii) keeping one dataset only, *i.e.*, MIMIC-IV including both 12-Lead ECGs and texts. Based on our training strategies and these two compared strategies, we assessed the performance disparity across varied lead settings on PTB-XL datasets. The radar axes in the figure are log-normalized and use the same range for better visualization. **b.** Comparison of pretraining strategies under different lead modalities and dataset combinations. Ablation models pretrained with (i) MIMIC-III PPG-only, (ii) MIMIC-III PPG+ECG, (iii) MIMIC-III PPG+ECG+Text, and (iv) the full heterogeneous pretraining set were compared across five PPG-related tasks. Performance values were min-max normalized within each task for clearer comparison.

C 1.2 — Quality of data and interpretation of results 2/2

Furthermore, to substantiate claims of a robust ECG foundation model, the authors should compare their approach against a few other existing ECG-only foundation models, rather than limiting the comparison to a single ECG-FM model. The discussion and conclusions could also use more detail regarding how this model performs compared to other foundational models. It is novel to try and combine features from various modalities, and this should be commended, but there is certainly room for improvement for downstream results.

Reply: Many thanks for these insightful comments. We have revised accordingly in terms of the following two aspects.

a. More ECG-only foundation models.

In addition to the previously included ECG-FM, we have now added more ECG-only foundation models: ECGFounder [4] (which provides open-sourced models trained separately on 12-lead and 1-lead ECGs), D-BETA [5] (specific to 12-lead ECG), and MERL [6] (also specific to 12-lead ECG).

For the specific Lead-II settings, similar to PPG comparison settings, we also compared with Chornos [2], Moment [3], and NormWear [1].

The results are shown in Figure 2 and Figure 1 (Figure 4 in the main paper, and Supplementary Figure S2 in the Supplementary.)

Across tasks, our model matches or outperforms existing ECG-only foundation models, while uniquely supporting flexible-lead configurations. Importantly, most existing ECG-only foundation models require a one-model-per-channel setting [4], which substantially limits their ability to transfer seamlessly between different lead configurations. By contrast, our framework naturally accommodates both 12-lead and fewer-lead inputs without retraining, providing a clear practical advantage.

b. Expanded analysis, discussion and opportunities.

We noticed the outstanding performance of ECGFounder [4] in certain cases, such as VTaC based False Alarm Prediction Figure 2b, and added discussion as below,

Main paper Page 10 “we observed particularly strong performance from ECGFounder [4] on the VTaC Lead-II setting, likely due to its pretraining on over 10 million ECG recordings. ”

Supplementary S3.2 “Across both settings, ECGFounder exhibited the strongest overall performance among existing foundation models, likely benefiting from extensive pretraining on large-scale ECG data and diagnostic label association during pretraining [4]. Future work may focus on incorporating diagnostic-specific priors and expanding pretraining to even larger clinical biosignal corpora, which may further narrow the gap with task-optimized foundation models such as ECGFounder. ”

To answer the above C 1.1C 1.2, we presented a detailed list of the foundation models we compared, and whether they are domain specific, and whether they support flexible-lead settings, the dataset size, in the Supplementary Material.

Relevant contents and table summary of these foundation models are as below:

Main paper Page 9 “Furthermore, we compared CSFM embeddings against those extracted from general time series models and from dedicated ECG/PPG foundation models. These are designed for time-series or physiological signals, including Chronos [2], Moment [3], NormWear [1], PaPaGei [7], ECG-FM [8], ECGFounder [4], D-BETA [5], and MERL [6]. Each model differs in modality compatibility and signal dimensionality.”

Supplementary S2.2 “**ECG-FM [8]** is a self-supervised foundation model pretrained on large-scale diagnostic ECGs using masked signal modeling. We employed the official weights pretrained on MIMIC-IV-ECG (<https://github.com/bowang-lab/ECG-FM>) and removed the final classification head for

feature extraction. Segment-level embeddings were derived by mean-pooling the final hidden layer across temporal positions. This model was directly applied to 12-lead ECGs.

“PaPaGei [7] is a foundation model pretrained on PPG signals from the VitalDB, MIMIC-III, and MESA [9] datasets. We followed the official tutorial (<https://github.com/nokia-bell-labs/papagei-foundation-model>) to process each 10-second PPG segment and extracted embeddings using the papagei_s version.

...”

Table 1: **Overview of foundation models utilized in this study.** Their originally supporting data type, pretrained ECG/PPG datasets, modality used in this work, and embedding dimensions are listed.

Model	Original Supporting Data Type	Pretrained ECG/PPG Datasets (if applicable)	Modality Used in This Study	Embedding Dim.
Chronos [2]	General univariate time series	–	Lead-II ECG, PPG (univariate)	768
Moment [3]	General multivariate time series	–	Lead-II ECG, PPG, ECG-PPG	1024
NormWear [1]	Wearable multivariate physiological data	Multiple physiological datasets (ECG, PPG, GSR, etc.)	Lead-II ECG, PPG, Lead-II ECG+PPG	768
PaPaGei [7]	PPG	VitalDB, MIMIC-III, MESA	PPG	512
ECG-FM [8]	12-Lead ECG	MIMIC-IV-ECG	12-lead ECG	768
ECGFounder [4]	12-Lead ECG, Single-Lead ECG (two versions)	Harvard-Emory	12-lead, single-lead ECG	1024
D-BETA [5]	12-Lead ECG	MIMIC-IV-ECG	12-lead ECG	768
MERL [6]	12-Lead ECG	MIMIC-IV-ECG	12-lead ECG	192

C 1.3 — Lack of context 1/2

Although it is understood that this is not a clinical journal, the discussion still could do a lot more to elaborate on the clinical significance of this model. That is, how could it be adapted by other groups, and what type of clinical impact could it have? Furthermore, there is little in the way of characterization of the datasets in terms of clinical and demographic factors that are important for real-world performance and furthermore could shed light on potential biases inherent in the embeddings. In addition, the Reporting Summary is devoid of pertinent information regarding demographic factors concerning the patient data that was used.

Reply: We thank the reviewers for this insightful point. We have accordingly made revisions to elaborate on the potential clinical impact of CSFM, as well as detailed characteristics of the datasets utilised.

a. Demographic characteristics.

Specifically, we have added a more detailed characterization of the datasets in the Table below (the Supplementary Material Table S2) and refer to these details where relevant in the main text. This includes available demographic factors such as age distribution, sex distribution, and geographic coverage. We have also updated the reporting summary to reflect such addition.

Table 2: **Dataset country or region, acquisition scenario, and population demographics.** Age is reported as median [interquartile range]; sex distribution is reported as the percentage of male participants. Statistics are calculated over all segments.

Dataset	Country/Region	Scenario	# Individuals	Age (years)	Sex (male, %)
MIMIC-III-WDB [†]	US	ICU	7,425	63 [52, 75]	56.0
MIMIC-IV-ECG [†]	US	Hospital / Clinical	160,821	66 [54, 77]	51.1
CODE-Full	Brazil	Primary Care / Telehealth	1,558,748	54 [41, 67]	39.7
PTB-XL	Germany	Hospital / Clinical	18,562	62 [50, 72]	52.1
SimBand	US	Simulated Ambulatory	40	-	-
CinC17	US	Ambulatory / Home	8,528	-	-
VitalDB	South Korea	Operating Room	1,409	61 [51, 70]	42.3
VTaC	US	ICU	2,147	-	-

[†] Ages above 89 years are truncated in MIMIC-III and MIMIC-IV in accordance with database de-identification policy.

b. Clarifying and emphasizing clinical impact.

Furthermore, we have revised the Discussion section, as well as the Results part throughout, to better underscore the clinical implications of our work. We would like to take this opportunity to highlight these revisions here.

The clinical impact of CSFM is supported by four main pillars, reflected in the corresponding Results subsections:

1. **Generalization capability across diverse healthcare tasks**, demonstrating the potential of CSFM to support a wide spectrum of diagnostic and monitoring applications beyond task-specific training.
2. **Versatility across modalities/channel settings**, enabling utility under varied lead configurations, signal types (ECG/PPG), and device conditions, which is an essential feature for real-world deployment in heterogeneous care environments.
3. **Generative capability from resource-limited to resource-intensive settings**, illustrating how CSFM can reconstruct or augment physiological signals, thereby enhancing data availability and accessibility in low-resource healthcare systems.

4. **Efficacy of embedding extraction**, where the learned representations provide reliable, transferable features for diverse downstream applications. This is particularly valuable for the current biosignal community, which lacks a standard toolbox capable of accommodating diverse data collection settings. CSFM thus offers an effective and unified feature extraction framework for different cardiac biosignals.

Overall in the Discussion, we emphasises the clinical impact

“ Our results demonstrate that CSFM robustly generalizes across a wide range of clinical scenarios, devices, and input configurations. Its generalizability across tasks, versatility across signals and modalities, generative capability spanning resource-limited to resource-intensive settings, and efficacy of embedding extraction, collectively help bridge the current fragmentation of digital health systems, where models are often confined by modality-specific silos and narrow task optimization. ”

Also, in line with C E.2, we have updated the the background and context of the work.

“Cardiovascular diseases remain a major contributor to the global burden of healthcare, highlighting the importance of accurate and scalable methods for cardiac monitoring. Cardiac biosignals, most notably electrocardiograms (ECG) and photoplethysmograms (PPG), are essential for diagnosing, preventing, and managing cardiovascular conditions across clinical and home settings. However, their acquisition varies substantially across scenarios and devices, while existing analytical models often rely on homogeneous datasets and static bespoke models, limiting their robustness and generalizability in diverse real-world contexts.”

c. Acknowledgement of potential bias.

In line with the clinical impact, we acknowledge that some biases may exist, although explicitly mitigating them was not the primary focus of this work, and we now clearly note this.

Supplementary Section S1.1 “We note that, despite the diversity of data sources, potential geographic or demographic biases may still exist, and the model’s generalizability to underrepresented populations should not be assumed without further validation.”

C 1.4 — Lack of context 2/2

In addition, the paper certainly utilizes many different datasets, as summarized in Figure 1, which can be seen as a strength. However, it is difficult while reading the main text to keep track of the number of classes, the type of dataset, and so on for all the different results, and so it is difficult to assess how successful the downstream results really are. This lack of clarity greatly hampers the potential impact of the results.

Reply: Many thanks, this is a very good point. We tried to clarify this from the following three aspects:

a. Detailed dataset characteristics and clear task mapping.

We have added more detailed dataset descriptions, including the modality (ECG, PPG, text), number of samples, and task type (classification or regression), as well as the class ratio where relevant.

A comprehensive summary table has been added below (Table S1 Supplementary Materials) and referenced in the main text.

Table 3: **Dataset characteristics, signal modalities, and associated downstream tasks.** DIR: Demographic Information Recognition; CDD: Cardiovascular Disease Diagnosis; VSM: Vital Sign Measurement; COP: Clinical Outcome Prediction; QA: ECG Question Answering.

Dataset	ECG	PPG	Text	# Segments	# Individuals	DIR	CDD	VSM	COP	QA
Pretraining	MIMIC-III-WDB [†]	Lead II	✓	Partial	270,562	7,425	–	–	–	–
	MIMIC-IV-ECG [†]	12 Leads	✗	✓	787,677	160,821	–	–	–	–
	CODE-Full	12 Leads	✗	✗	1,558,748	1,558,748	–	–	–	–
Downstream	PTB-XL	12 Leads	✗	✗	21,388	18,562	–	✓	–	–
	SimBand	–	✓	✗	7,590	40	–	✓	–	–
	CinC17	Wearable	✗	✗	8,528	8,528	–	✓	–	–
	VitalDB	Lead II	✓	✗	574,800	1,409	✓	–	✓	–
	VTaC	Lead II	✓	✗	4,613	2,147	–	–	–	✓
	CODE-15	12 Leads	✗	✗	233,647	233,647	–	–	–	✓
	ECG-QA*	12 Leads	✗	✓	231,536	–	–	–	–	✓

*A version modified from PTB-XL.

[†]MIMIC IV and MIMIC III may include overlapping individuals but are de-identified and cannot be linked.

On the other hand, we have revised Figure 1-c (originally Figure 2b) as below, to clarify the mapping between downstream tasks and datasets, for better understanding of the utilisation of each dataset.

Figure 5: Updated datasets for better mapping between tasks and datasets

b. Clear and consistent figure annotations.

In line with C 1.11, we now explicitly annotate each downstream result with the corresponding dataset and modality used, especially for Figure 6 (Figure 2, originally Figure 3 in the main paper).

Figure 6: Overall performance across different healthcare scenarios, validated on corresponding downstream datasets, separately.

c. Consistent result referencing.

For all other result figures, we have added clear visual indicators (e.g., dashed lines) to align with Figures 2 and 3 of the main paper, ensuring consistent presentation of dataset–task associations across all experiments. It is demonstrated in Figure 2 and Figure 1 (Figures 4, S2 in main paper). It is also clearly mentioned in the caption, for instance, “reference lines indicate the best-performing CSFM-based and non-CSFM-based methods, obtained either via fine-tuning or direct training, as reported in Figures 2 and 3.”

C 1.5 — Classification metrics: Macro-F1 emphasizes correct classification (TP or TN) for each class individually while under-penalizing misclassifications (FP, FN). This makes it less reliable in imbalanced settings. However, metrics like the Matthews Correlation coefficient (MCC), which account for all TP, TN, FP, and FN simultaneously, provide a more balanced evaluation. It is therefore suggested to report MCC scores alongside or instead of macro-F1.

Reply: Thank you, we have included MCC scores in the Supplementary alongside Macro-F1 for completeness.

The results from Table S5 are as below,

Table 4: **Performance comparison of different methods across three ECG datasets (PTB-XL, CinC17, and SimBand).** Performance was measured using Macro-F1 and MCC metrics (mean [95% CI] across three random seeds). XGB prediction based on domain features is also listed for benchmarking.

Method	SimBand		CinC17		PTB-XL	
	Macro-F1	MCC	Macro-F1	MCC	Macro-F1	MCC
Domain Feature	0.328 [0.308, 0.348]	0.360 [0.311, 0.410]	0.496 [0.470, 0.523]	0.393 [0.363, 0.424]	0.258 [0.256, 0.260]	0.244 [0.238, 0.251]
BiLSTM	0.332 [0.302, 0.364]	0.332 [0.306, 0.358]	0.588 [0.540, 0.636]	0.455 [0.370, 0.540]	0.288 [0.260, 0.315]	0.293 [0.260, 0.327]
ResNet1d18	0.357 [0.324, 0.390]	0.435 [0.338, 0.532]	0.606 [0.557, 0.655]	0.484 [0.386, 0.583]	0.324 [0.307, 0.342]	0.323 [0.303, 0.344]
ResNet1d34	0.334 [0.308, 0.360]	0.329 [0.227, 0.431]	0.634 [0.558, 0.710]	0.557 [0.451, 0.662]	0.328 [0.296, 0.361]	0.325 [0.278, 0.373]
ResNet1d50	0.347 [0.341, 0.353]	0.354 [0.278, 0.429]	0.561 [0.495, 0.627]	0.496 [0.439, 0.553]	0.322 [0.317, 0.328]	0.319 [0.306, 0.332]
ResNet1d101	0.351 [0.297, 0.405]	0.378 [0.147, 0.609]	0.592 [0.508, 0.675]	0.518 [0.465, 0.572]	0.326 [0.304, 0.347]	0.321 [0.300, 0.341]
Inception1d	0.344 [0.323, 0.365]	0.337 [0.262, 0.411]	0.601 [0.517, 0.684]	0.462 [0.386, 0.537]	0.323 [0.285, 0.361]	0.317 [0.284, 0.350]
MSDNN	0.337 [0.315, 0.360]	0.321 [0.199, 0.442]	0.618 [0.570, 0.666]	0.537 [0.466, 0.608]	0.294 [0.268, 0.320]	0.283 [0.259, 0.307]
CSFM-Tiny	0.354 [0.338, 0.370]	0.384 [0.346, 0.421]	0.656 [0.646, 0.667]	0.575 [0.542, 0.608]	0.342 [0.309, 0.374]	0.342 [0.313, 0.372]
CSFM-Base	0.354 [0.314, 0.395]	0.417 [0.283, 0.550]	0.655 [0.617, 0.694]	0.631 [0.609, 0.653]	0.357 [0.338, 0.377]	0.373 [0.364, 0.383]
CSFM-Large	0.398 [0.279, 0.516]	0.439 [0.405, 0.473]	0.677 [0.656, 0.698]	0.620 [0.568, 0.671]	0.338 [0.314, 0.362]	0.412 [0.396, 0.429]

Discussion in Supplementary Section S3.1 “Among the baseline models, ResNet1d18-101 and MS-DNN achieve moderate performance, with Macro-MCC values typically ranging between 0.28 and 0.55 across datasets. In contrast, the proposed CSFM models consistently outperform these baselines.

Specifically, CSFM-Large achieves the highest MCC on SimBand (0.439 [0.405, 0.473]) and PTB-XL (0.412 [0.396, 0.429]), indicating stronger generalization across both wearable and clinical ECG domains. On the CinC17 dataset, CSFM-Base attains the best MCC (0.631 [0.609, 0.653]), reflecting stable discriminative capability across cardiac conditions. These findings align with the Macro-F1 results, demonstrating that larger CSFM variants provide consistent gains over conventional convolutional architectures.”

C 1.6 — Clinical Outcome Prediction: In the subsection of the results on clinical outcome prediction, the description of the results lacks sufficient detail regarding the underlying dataset. Other subsections provide this context, and the same should be done here for consistency and clarity.

Reply: Many thanks for the detailed look. We have revised the corresponding subsection as below,

Main Page 5 “In this scenario, we evaluated the predictive performance of our models by forecasting the likelihood of adverse events. ... The former utilized the VTaC dataset [10], while the latter employed the CODE-15 dataset [11], a publicly available subset of CODE-FULL. We split CODE-15 subject-wise (80% training, 10% validation, 10% testing) and VTaC based on its official split, and conducted binary classification for the outcome prediction on each dataset, respectively. It should be noted that CODE-15 is a public small version of CODE-Full [11], and in experimental settings, we ensured that no training subjects in CODE-Full are available in the validation/testing subset of CODE-15. These evaluations demonstrate the versatility of our approach across both immediate critical care and long-term risk stratification tasks. The receiver operating characteristic (ROC) curves, which illustrate the accuracy of our predictions, are presented in Figure 6d,e. Additional results from three random seeds are available in Figure 3a. Our results show that the AUC for the CSFM series reaches up to 0.844 for 1-year mortality prediction, compared to 0.816 for conventional deep learning methods trained from scratch. Additionally, the false alarm prediction on VTaC achieves superior performance relative to traditional approaches, with best CSFM reaching AUC of 0.967 against 0.946 for traditional methods.”

C 1.7 — Feature Extraction and Methods: In the Supplemental Information, a comprehensive list should be provided of the domain features extracted from both PPG and ECG signals. If an exhaustive listing is not feasible, at least report the major features used. Additionally, the hyperparameters applied to the three classifiers should be explicitly listed, as this information is essential for reproducibility.

Reply: Many thanks, we have now provided a list of all the features extracted from both PPG and ECG signals, as well as the hyperparameters applied to the three classifiers, to support reproducibility.

The added content is as below,

Supplementary S2.1 “We extracted commonly used domain-specific features to serve as a baseline for comparison with features learned by CSFM. For ECG, we computed 54 handcrafted features per channel, capturing both morphological characteristics and heart rate variability. Feature extraction was based on R-peak detection and waveform delineation using NeuroKit2. For multi-lead ECGs, features from individual channels were averaged to obtain a consolidated representation. For PPG, we used pyPPG to extract a total of 306 features reflecting pulse morphology and variability. All feature extraction scripts are included in the supplementary code. ”

“Following standard scaling of the input features, three types of classical machine learning models were applied to evaluate the discriminative quality of the extracted signal embeddings: logistic regression (`scikit-learn`), random forest (`scikit-learn`), and XGBoost (`xgboost` package). The logistic regression model used default settings with `max_iter=1000`. The random forest model was configured with `n_estimators=300` and the XGBoost model was configured with `n_estimators=300` and `max_depth=6`, and a learning rate of 0.05, while all other parameters remained at their default values. For regression tasks, the XGBoost regressor used the same configuration with `objective='reg:squarederror'` and `eval_metric='rmse'`. ”

On the other hand, we also investigate the relationship between physiological features and learned latent representations.

Table 5: **Comparison of handcrafted ECG and PPG feature groups.** All features are organized into three interpretable physiological categories. Representative examples are shown with feature names and short descriptions.

Feature Group	ECG Features	PPG Features
Interval-related features	Timing between cardiac events and intra-beat durations. Examples: [RR0] previous R–R interval, [t_PR] P–R interval, [t_QT] Q–T interval, [RR_m/1] average-to-current RR ratio.	Pulse-to-pulse intervals, widths, and fiducial timings. Examples: [Tpi] pulse onset-to-offset interval, [Tpp] peak-to-peak interval, [Tsp] systolic peak time, [Tsw50] systolic width at 50% amplitude.
Amplitude-related features	Waveform heights and voltage differences. Examples: [a_R] R-peak amplitude, [a_RS] R–S drop, [a_ST/QS] repolarization-to-depolarization ratio, [a_RS/QR] QRS symmetry index.	Waveform magnitudes, amplitude ratios, and area-based measures. Examples: [Asp] systolic peak amplitude, [Adn] dicrotic notch amplitude, [Adp] diastolic peak amplitude, [AUCsys] area under the systolic segment.
Other temporal / dynamic features	Beat-to-beat variability and signal quality. Examples: [SDNN] standard deviation of NN intervals, [RMSSD] root mean square of successive differences, [TINN] triangular HRV index, [ECG_SQI] signal quality index.	Waveform dynamics, arterial stiffness, and composite indices. Examples: [AI] augmentation index (Tp2–Tp1 difference / Asp), [RIp1] reflection index at p1, [RIp2] reflection index at p2, [IPAD] inflection-point area + normalized d-point amplitude.

Supplementary S3.6 “... we conducted a correlation analysis between the top domain-engineered features and the principal components (PCs) of the foundation-model embeddings. We trained an XG-Boost classifier on handcrafted domain features to identify the top ten most informative features for the downstream classification task, as shown in Figure 11. Feature importance was determined using gain-based importance scores averaged across trees. Meanwhile, the 768-dimensional embeddings from each foundation model (CSFM-Tiny, ECGFounder, or PaPaGei) were standardized and projected onto their first ten principal components (PCs) using Principal Component Analysis (PCA). These PCs capture the major directions of variance in the embedding space. For each dataset, we computed Spearman’s rank correlation coefficient between the ten selected domain features and the ten PCA components. The resulting 10×10 correlation matrices quantify how strongly each interpretable physiological variable aligns with individual embedding dimensions. Each cell in the heatmap corresponds to the correlation between a domain feature (row) and a PCA component (column).”

“Positive correlations (blue) indicate that a given PCA direction increases monotonically with a physiological measure (e.g., heart rate variability indices such as pNN50 or RMSSD), while negative correlations (red) suggest an inverse relationship. The magnitude of the correlation value reflects the strength of alignment between an interpretable domain feature and a latent axis of the embedding space.”

Figure 7: Spearman correlation between the top 10 most informative domain features and the first 10 principal components (PCs) of foundation-model embeddings. Blue indicates positive correlation, red indicates negative correlation. The analysis links interpretable physiological descriptors (rows) with latent representation axes (columns) for ECG (leftmost two) and PPG (rightmost two) modalities. $pNN20/pNN50$: percentage of successive RR differences $> 20 / 50$ ms; SDD : standard deviation of successive RR interval differences; $CVSD/CVNN/MCVNN$: coefficients of variation of RR intervals or their differences; $Prc20NN/Prc80NN$: 20th/80th percentiles of RR intervals; $MedianNN/MinNN$: central and minimal RR intervals; Adn : dicrotic-notch amplitude; Tpi : pulse interval between consecutive pulse onsets; $TPw10/TPw50/TPw75$: pulse-width durations at 10%, 50%, 75% amplitude; $Asp/Asp/Aoff$: systolic amplitude and its ratio to baseline; TPp : peak-to-peak interval; $TPw50/Tpi/TPw75/Tsp$: ratios of width to interval or systolic-time, capturing waveform symmetry and arterial stiffness; Tdp : diastolic-peak time.

C 1.8 — Language and Terminology Issues: The manuscript contains a few spelling and typographical errors (e.g., “biosignals” misspelled on page 14, “mean” incorrectly typed as “meas” on page 6 in the demographic information subsection). These should be corrected throughout. Furthermore, acronyms are not always defined prior to use (e.g., “MLP”, “CVD” appeared without prior definition). In some cases, acronyms are redefined multiple times or used ambiguously; for instance, “MAE” is defined multiple times as “mean absolute error” and again later used to denote “masked autoencoder”. This creates unnecessary confusion. Acronyms should be defined once, used consistently, and never overloaded with multiple meanings.

Reply: Tremendous thanks, this has been fixed. Meanwhile, we did another round of proofreading to ensure all acronyms are clearly and appropriately defined and used.

C 1.9 — Figure 1 - The coloring of Figure 1 is confusing. It took a few views to understand the various shades of blue/gray.

Reply: Thanks, we would like to thank the reviewers for pointing out this unintended ambiguity issue. Originally, different colors in the Sensing Devices block were meant to highlight the comprehensive coverage of cardiac biosignal modalities and to indicate devices in transition, from gold-standard measurement to continuous monitoring, and from hospital to home. To avoid potential confusion, we have revised the figure to more clearly convey this notion of “transition”.

Figure 8: Updated Figure 1a.

C 1.10 — Figure 2 - The color coding for panels (a) and (b) is too similar and may confuse readers. Distinct color schemes should be used. Likewise, in Figure 6, the colors representing different features are nearly indistinguishable, which makes interpretation difficult. Use clearly distinct colors or alternative visual encodings to improve readability.

Reply: Thank you very much! We have significantly revised all figures to make them more accessible. For your convenience, we have attached the specific figures mentioned, original Figure 2 (current Figure 1-b,c), and original Figure 6 (current Figure 4-a), as below.

Figure 9: Updated Figure 1b-c and Figure 4-a.

C 1.11 — Figure 3 - Figure labeling is inconsistent. While the first figure specifies classes, sample sizes, and signal types, subsequent figures omit such details. For consistency and clarity, we strongly recommend adopting a uniform style across all figures, and when possible, referencing the relevant tables or sections for this information in the captions.

- What is the class balance for Figure 3a? The Macro F1 Scores seem poor, especially for only four classes. Could the 44 classes be combined somehow? It may be too difficult of a task. Figure 3b needs to emphasize that VitalDB only uses a Lead II ECG when discussing the demographic recognition because otherwise the error/AUC is poor.

- Figure 3c is unclear whether the depicted waveform represents a single example trace or an average across waveform segments from the VitalDB dataset. Please clarify this in the caption.

Reply: Thank you very much, we have revised original Figure 3, now Figure 2 in terms of these specific points. Please refer to Figure 6 for more details.

In terms of the results in old Figure 3a, we acknowledge that the observed variation in macro-F1 performance is likely influenced by class imbalance (smallest 0.07%, largest 44.48%) within the dataset, as noted in Supplementary Section S1.2. Nevertheless, CSFM demonstrates superior or comparable performance compared to others consistently.

We have tried to merge 44 classes into 5 super classes, following PTB-XL settings [12], and provided additional results in the Supplementary.

Supplementary Section S3.2 “PTB-XL can also be grouped into 5 superclass categories, *i.e.*, Normal (NORM), Myocardial Infarction (MI), ST/T Change (STTC), Hypertrophy (HYP), and Conduction Disturbance (CD)), representing broader diagnostic groups. To further examine model generalization under different label granularities, we conducted additional experiments on both the 5-class superclass and the full 44-class diagnostic settings, based on XGBoost.

As shown in Supplementary Table 6, the trends in performance between the two settings are consistent. When evaluated under the coarser 5-class configuration, all models achieved substantially higher F1 and MCC scores, reflecting the reduced label complexity. Across both configurations, CSFM variants achieved competitive or superior performance compared with existing foundation models, particularly in the 44-class setting, where CSFM-Base attained the highest MCC (0.330 [0.315, 0.346]) and CSFM-Large achieved the best F1 (0.331 [0.316, 0.346]). These findings highlight the robustness of CSFM representations in multi-label ECG classification tasks of varying difficulty.

Across both settings, ECGFounder exhibited the strongest overall performance among existing foundation models, likely benefiting from extensive pretraining on large-scale ECG data and diagnostic label association during pretraining [4]. Future work may focus on incorporating diagnostic-specific priors and expanding pretraining to even larger clinical biosignal corpora, which may further narrow the gap with task-optimized foundation models such as ECGFounder. ”

C 1.12 — Table 2 - Some context of what a good metric value would be for the dataset and classes would be helpful.

Reply: Thank you, we have emphasized this supplementary and linked it with the main context.

Supplementary Section S1.4 “The evaluation metrics for the three groups of downstream tasks are summarized as follows.

Binary classification. For binary tasks, the primary evaluation metric was the area under the receiver operating characteristic curve (AUC), as it provides a threshold-independent measure of discriminative performance.

Table 6: **Comparison between 5-class superclass and 44-class PTB-XL diagnostic settings.** All values are mean [95% CI] across three random seeds. **Bold** indicates best overall performance.

Model	Superclass (5 Classes)		Main Paper Settings (44 Classes)	
	Macro-F1	MCC	Macro-F1	MCC
Domain Feature	0.555 [0.548, 0.562]	0.393 [0.387, 0.399]	0.258 [0.256, 0.260]	0.244 [0.238, 0.251]
ECG-FM	0.537 [0.529, 0.545]	0.392 [0.383, 0.401]	0.267 [0.262, 0.271]	0.254 [0.248, 0.259]
D-BETA	0.563 [0.555, 0.572]	0.435 [0.425, 0.445]	0.231 [0.223, 0.239]	0.235 [0.206, 0.263]
MERL	0.591 [0.586, 0.597]	0.445 [0.440, 0.451]	0.273 [0.264, 0.282]	0.266 [0.258, 0.274]
ECGFounder	0.618 [0.610, 0.626]	0.505 [0.497, 0.513]	0.310 [0.304, 0.316]	0.310 [0.297, 0.323]
CSFM-Tiny	0.604 [0.602, 0.606]	0.470 [0.463, 0.477]	0.309 [0.287, 0.330]	0.304 [0.278, 0.329]
CSFM-Base	0.596 [0.590, 0.602]	0.469 [0.461, 0.477]	0.327 [0.321, 0.333]	0.330 [0.315, 0.346]
CSFM-Large	0.604 [0.603, 0.606]	0.480 [0.474, 0.487]	0.331 [0.316, 0.346]	0.323 [0.308–0.339]

Multi-class and multi-label classification. For multi-class and multi-label tasks, the primary metric was the macro-averaged F1-score, which equally weights all classes and is less affected by label imbalance. For completeness, Matthews correlation coefficient (MCC) values are also reported in the Supplementary Table 8 for complementary evaluation. In particular, the ECG-QA task is formulated as a multi-label classification problem, where the model selects the correct answer(s) from a predefined set of candidate options. When computing the metrics, false positives outside the given candidates are not considered. Therefore, a modified version of the F1-score [13] was used to better reflect task-specific accuracy.

Regression. For regression tasks, such as vital sign estimation, the mean absolute error (MAE) was used as the primary evaluation metric, as it is more interpretable and less sensitive to outliers than the root mean squared error (RMSE). The RMSE was additionally reported, in some cases, to provide complementary assessments.

All metrics were computed on held-out test sets using implementations from the `scikit-learn` library. ”

On the other hand, in line with C 1.5, as suggested, we have also reported MCC in Table 8 (Supplementary Table S5).

C 1.13 — Table 3 - The metrics from the reconstruction from either PPG and a single lead are not great. Because AF can be distinguished with the human eye, could the authors speculate why the AUCs are not higher?

Reply: Thank you very much, this may suggest the gap between the real ECG and PPG-generated synthetic ECG data. We tried to address this from the following two parts, in line with Supplementary S3.6:

a. Why train-on-real test-on-synthetic ECG is that high:

This likely reflects the intrinsic gap between ECG and PPG modalities.

“PPG signals (and occasionally ECG) are inherently noisy, and the use of a one-to-one reconstruction objective, such as mean squared error (MSE), can lead the model to learn overly smoothed waveforms that fail to capture real-world artifacts and the diverse physiological associations between PPG and ECG. Previous findings in the computer vision domain [14] have shown that MSE-based generative optimization tends to produce over-smoothed results with limited diversity, motivating the development of more advanced generative approaches such as adversarial, diffusion, or variational models.”

Representative examples are shown in Figure 10b (Supplementary Figure S5b), where the generated ECG signals appear smoother than the real ECG traces, which contain more high-frequency fluctuations and physiological noise.

b. Why train-on-synthetic test-on-real is lower, and there are large gap:

This is most likely caused by the diversity mismatch.

“Although CinC17 and SimBand have similar segment sizes of around 8k, CinC17 contains 8,528 subjects, whereas SimBand includes only 40 individuals, thus constraining the diversity and representativeness of the synthetic domain. We introduced data augmentation strategies during synthetic pretraining, specifically temporal dropout, random scaling, and Gaussian noise, as shown in Figure 10a). These augmentations yielded measurable improvements and partially reduced the performance gap, although some of these gains may stem from the augmentations themselves.”

Figure 10: **Cross-modality reconstruction and augmentation results.** **a.** PPG to ECG Reconstruction. (1) The reconstruction was performed on VitalDB, and the waveform reconstruction performance was reported on the held-out test set of VitalDB. (2) Subsequently, we applied the adapted model to the original SimBand dataset to generate synthetic Lead-II ECG waveforms. To comprehensively test the quality of generated ECG waveforms, we conducted two experimental settings: train on synthetic ECG from SimBand (normal versus AF), and test on real ECG on CinC17 (normal versus AF), and vice versa. The performance is reported using F1 and AUC. **Best** values are in bold, and second best are underlined. **b.** Examples of real and synthetic ECGs belonging to normal and AF.

a

Methods	Waveform Reconstruction		Train-Real Test-Synthetic		Train-Syn (w/o Aug) Test-Real		Train-Syn (w Aug) Test-Real	
	MAE	RMSE	F1	AUC	F1	AUC	F1	AUC
UNet1d	0.608	0.964	0.427	0.591	0.276	0.558	0.453	0.534
BiLSTM	0.607	0.953	0.536	0.648	0.340	0.622	0.590	0.677
AutoEncoder[15]	0.585	0.927	0.442	0.784	0.353	0.600	0.420	0.607
CSFM-Tiny	0.532	0.863	0.690	0.812	0.365	0.669	0.570	0.749
CSFM-Base	0.524	0.852	0.632	0.815	0.364	0.690	0.590	0.703
CSFM-Large	0.516	0.840	0.692	0.820	0.353	0.688	0.585	0.731

b

Reviewer 4

C 4.1 — The model is trained mostly on MIMIC datasets where data is from US based ICU settings, there could be limited global population and unclear generalization to outpatient care.

Reply: Thanks for raising this point. We would like to address based on the following two aspects.

a. Clarifying the geographic differences of the datasets utilized.

In line with C 1.3, in supplementary materials, we have added a demographic summary of all datasets, including age, gender, country, and collection scenarios. The Table is shown as below (Supplementary Table S2)

Supplementary Section S1.1 “These datasets cover a wide spectrum of acquisition settings, including intensive care units, operating rooms, primary care, telehealth, and home-based monitoring. Pretraining was conducted using major datasets collected primarily in the United States and Brazil, while downstream evaluations were performed across datasets from multiple countries worldwide.”

Table 7: **Dataset country or region, acquisition scenario, and population demographics.** Age is reported as median [interquartile range]; sex distribution is reported as the percentage of male participants. Statistics are calculated over all segments.

Dataset	Country/Region	Scenario	# Individuals	Age (years)	Sex (male, %)
MIMIC-III-WDB [†]	US	ICU	7,425	63 [52, 75]	56.0
MIMIC-IV-ECG [†]	US	Hospital / Clinical	160,821	66 [54, 77]	51.1
CODE-Full	Brazil	Primary Care / Telehealth	1,558,748	54 [41, 67]	39.7
PTB-XL	Germany	Hospital / Clinical	18,562	62 [50, 72]	52.1
SimBand	US	Simulated Ambulatory	40	-	-
CinC17	US	Ambulatory / Home	8,528	-	-
VitalDB	South Korea	Operating Room	1,409	61 [51, 70]	42.3
VTaC	US	ICU	2,147	-	-

[†] Ages above 89 years are truncated in MIMIC-III and MIMIC-IV in accordance with database de-identification policy.

We have also clarified dataset origins and their respective healthcare settings in the caption of Figure 1b,c of the main paper:

“**b.** Pretraining integrates data from MIMIC-III-WDB (US), MIMIC-IV-ECG (US), and CODE-FULL (Brazil), comprising approximately 1.7 million heterogeneous cardiac-related biosignals and texts. Distributions by dataset source and signal modality are shown. **c.** Downstream evaluation covers five representative tasks, including cardiac disease diagnosis (CDD), demographic information recognition (DIR), vital sign measurement (VSM), clinical outcome prediction (COP), and ECG-based question answering (QA), using datasets such as CinC17 (US), PTB-XL (Germany), SimBand (US), VTaC (US), CODE-15 (Brazil), and VitalDB (South Korea), spanning diverse healthcare settings and populations. Detailed dataset descriptions and statistics are provided in Supplementary Table S1 and Section S1.”

b. Limitation and potential future work.

We agree with the reviewer that, despite including data from multiple countries and healthcare environments, geographic and demographic biases may still exist.

Supplementary Section S1.1 “We note that, despite the diversity of data sources, potential geographic or demographic biases may still exist, and the model’s generalizability to underrepresented populations should not be assumed without further validation.”

At the same time, we acknowledge that further validation on larger global population datasets is necessary to fully establish generalizability. We have expanded the Discussion to highlight this as an important direction for future work.

Main Page 10 “Moreover, expanding training to include larger and more diverse datasets would further enhance robustness in certain settings, as well as uncover potential biases.”

C 4.2 — The evaluation was done comparing all deep learning models, with transformer over other sequential models for this specific domain, would be beneficial to compare traditional sequential models and add some discussion of concurrent foundation model developments in healthcare.

Reply: Many thanks for the insightful point.

a. Expanded experiments with traditional and domain-feature-based models.

In addition to transformer-based architectures, we have incorporated traditional sequential models, namely BiLSTM, for comparison, as reported in the Supplementary Table 8.

On the other hand, while traditional deep models remain largely black-box and less interpretable, domain-feature-based models (e.g., XGBoost trained on handcrafted ECG/PPG features) provide interpretable yet less powerful alternatives. We also provided the result in Table 8. We have included such domain-feature baselines across downstream tasks, in the main paper Figure 4, and Supplementary Figure S2. We also leverage such interpretability to uncover the “black box” of foundation models as well, as illustrated in c.

Table 8: **Performance comparison of different methods across three ECG datasets (PTB-XL, CinC17, and SimBand).** Performance was measured using Macro-F1 and MCC metrics (mean [95% CI] across three random seeds). XGB prediction based on domain features is also listed for benchmarking.

Method	SimBand		CinC17		PTB-XL	
	Macro-F1	MCC	Macro-F1	MCC	Macro-F1	MCC
Domain Feature	0.328 [0.308, 0.348]	0.360 [0.311, 0.410]	0.496 [0.470, 0.523]	0.393 [0.363, 0.424]	0.258 [0.256, 0.260]	0.244 [0.238, 0.251]
BiLSTM	0.332 [0.302, 0.364]	0.332 [0.306, 0.358]	0.588 [0.540, 0.636]	0.455 [0.370, 0.540]	0.288 [0.260, 0.315]	0.293 [0.260, 0.327]
ResNet1d18	0.357 [0.324, 0.390]	0.435 [0.338, 0.532]	0.606 [0.557, 0.655]	0.484 [0.386, 0.583]	0.324 [0.307, 0.342]	0.323 [0.303, 0.344]
ResNet1d34	0.334 [0.308, 0.360]	0.329 [0.227, 0.431]	0.634 [0.558, 0.710]	0.557 [0.451, 0.662]	0.328 [0.296, 0.361]	0.325 [0.278, 0.373]
ResNet1d50	0.347 [0.341, 0.353]	0.354 [0.278, 0.429]	0.561 [0.495, 0.627]	0.496 [0.439, 0.553]	0.322 [0.317, 0.328]	0.319 [0.306, 0.332]
ResNet1d101	0.351 [0.297, 0.405]	0.378 [0.147, 0.609]	0.592 [0.508, 0.675]	0.518 [0.465, 0.572]	0.326 [0.304, 0.347]	0.321 [0.300, 0.341]
Inception1d	0.344 [0.323, 0.365]	0.337 [0.262, 0.411]	0.601 [0.517, 0.684]	0.462 [0.386, 0.537]	0.323 [0.285, 0.361]	0.317 [0.284, 0.350]
MSDNN	0.337 [0.315, 0.360]	0.321 [0.199, 0.442]	0.618 [0.570, 0.666]	0.537 [0.466, 0.608]	0.294 [0.268, 0.320]	0.283 [0.259, 0.307]
CSFM-Tiny	0.354 [0.338, 0.370]	0.384 [0.346, 0.421]	0.656 [0.646, 0.667]	0.575 [0.542, 0.608]	0.342 [0.309, 0.374]	0.342 [0.313, 0.372]
CSFM-Base	0.354 [0.314, 0.395]	0.417 [0.283, 0.550]	0.655 [0.617, 0.694]	0.631 [0.609, 0.653]	0.357 [0.338, 0.377]	0.373 [0.364, 0.383]
CSFM-Large	0.398 [0.279, 0.516]	0.439 [0.405, 0.473]	0.677 [0.656, 0.698]	0.620 [0.568, 0.671]	0.338 [0.314, 0.362]	0.412 [0.396, 0.429]

b. Expanded comparison and discussion against concurrent foundation model development.

We have expanded our analysis across more foundation models, for specific ECG (ECG-FM [8], ECGFounder [4], D-BETA [5], and MERL [6]), PPG (PaPaGei [7]), physiological signals [1], or for general time series (Chronos [2], Moment [3]).

Main Page 9 “Furthermore, we compared CSFM embeddings against those extracted from general time series models and from dedicated ECG/PPG foundation models. These are designed for time-series or physiological signals, including Chronos [2], Moment [3], NormWear [1], PaPaGei [7], ECG-FM [8], ECGFounder [4], D-BETA [5], and MERL [6]. Each model differs in modality compatibility and signal dimensionality. Brief descriptions and implementation details for these models are summarized

in Supplementary Section S2.2 and Table S4. Their performance was assessed across multiple datasets and tasks, including PTB-XL, SimBand, CinC17, and VitalDB, evaluated using an XGBoost classifier. As shown in Figure 2b–d, modality-specific foundation models generally surpass general-purpose time-series models, and we observed particularly strong performance from ECGFounder [4] on the VTaC Lead-II setting, likely due to its pretraining on over 10 million ECG recordings. Additional results for VitalDB-based BMI and gender prediction are provided in the Supplementary Material (Section S3.3).”

These foundation models are discussed in detail in the supplementary (Supplementary S3.3), and a summary table (Supplementary Table S4) is also provided below,

Supplementary S2.2 “**ECG-FM [8]** is a self-supervised foundation model pretrained on large-scale diagnostic ECGs using masked signal modeling. We employed the official weights pretrained on MIMIC-IV-ECG (<https://github.com/bowang-lab/ECG-FM>) and removed the final classification head for feature extraction. Segment-level embeddings were derived by mean-pooling the final hidden layer across temporal positions. This model was directly applied to 12-lead ECGs.

“**PaPaGei [7]** is a foundation model pretrained on PPG signals from the VitalDB, MIMIC-III, and MESA [9] datasets. We followed the official tutorial (<https://github.com/nokia-bell-labs/papagei-foundation-model>) to process each 10-second PPG segment and extracted embeddings using the papagei_s version.

Table 9: **Overview of foundation models utilized in this study.** Their originally supporting data type, pretrained ECG/PPG datasets, modality used in this work, and embedding dimensions are listed.

Model	Original Supporting Data Type	Pretrained ECG/PPG Datasets (if applicable)	Modality Used in This Study	Embedding Dim.
Chronos [2]	General univariate time series	–	Lead-II ECG, PPG (univariate)	768
Moment [3]	General multivariate time series	–	Lead-II ECG, PPG, ECG-PPG	1024
NormWear [1]	Wearable multivariate physiological data	Multiple physiological datasets (ECG, PPG, GSR, etc.)	Lead-II ECG, PPG, Lead-II ECG+PPG	768
PaPaGei [7]	PPG	VitalDB, MIMIC-III, MESA	PPG	512
ECG-FM [8]	12-Lead ECG	MIMIC-IV-ECG	12-lead ECG	768
ECGFounder [4]	12-Lead ECG, Single-Lead ECG (two versions)	Harvard-Emory	12-lead, single-lead ECG	1024
D-BETA [5]	12-Lead ECG	MIMIC-IV-ECG	12-lead ECG	768
MERL [6]	12-Lead ECG	MIMIC-IV-ECG	12-lead ECG	192

c. Expanded analysis of foundation models and domain features. Furthermore, we also investigated the relationship between these domain features and foundation model-derived embeddings,

Supplementary Section S3.4 “... we conducted a correlation analysis between the top domain-engineered features and the principal components (PCs) of the foundation-model embeddings. We trained an XGBoost classifier on handcrafted domain features to identify the top ten most informative features for the downstream classification task, as shown in Figure 11. Feature importance was determined using gain-based importance scores averaged across trees. Meanwhile, the 768-dimensional embeddings from each foundation model (CSFM-Tiny, ECGFounder, or PaPaGei) were standardized and projected onto their first ten principal components (PCs) using Principal Component Analysis (PCA). These PCs capture the major directions of variance in the embedding space. For each dataset, we computed Spearman’s rank correlation coefficient between the ten selected domain features and the ten PCA components. The resulting 10×10 correlation matrices quantify how strongly each interpretable physiological variable aligns with individual embedding dimensions. Each cell in the heatmap corresponds to the correlation between a domain feature (row) and a PCA component (column).”

“Positive correlations (blue) indicate that a given PCA direction increases monotonically with a physiological measure (e.g., heart rate variability indices such as pNN50 or RMSSD), while negative correlations (red) suggest an inverse relationship. The magnitude of the correlation value reflects the

strength of alignment between an interpretable domain feature and a latent axis of the embedding space.”

Figure 11: Spearman correlation between the top 10 most informative domain features and the first 10 principal components (PCs) of foundation-model embeddings. Blue indicates positive correlation, red indicates negative correlation. The analysis links interpretable physiological descriptors (rows) with latent representation axes (columns) for ECG (leftmost two) and PPG (rightmost two) modalities. $pNN20/pNN50$: percentage of successive RR differences $> 20 / 50$ ms; SDD : standard deviation of successive RR interval differences; $CVSD/CVNN/MCVNN$: coefficients of variation of RR intervals or their differences; $Prc20NN/Prc80NN$: 20th/80th percentiles of RR intervals; $MedianNN/MinNN$: central and minimal RR intervals; Adn : dicrotic-notch amplitude; Tpi : pulse interval between consecutive pulse onsets; $TPw10/TPw50/TPw75$: pulse-width durations at 10%, 50%, 75% amplitude; $Asp/Asp/Aoff$: systolic amplitude and its ratio to baseline; Tpp : peak-to-peak interval; $TPw50/Tpi/TPw75/Tsp$: ratios of width to interval or systolic-time, capturing waveform symmetry and arterial stiffness; Tdp : diastolic-peak time.

C 4.3 — CSFM-Large sometimes performs worse than CSFM-Base, e.g., figure 3, maybe scaling up the parameters are not optimal

Reply: Thank you very much. Yes, we agree that in some cases, CSFM-Large performs slightly worse than CSFM-Base.

During pretraining, we set up validation to monitor training progress. Main Page 14 “The dataset was split into training (80%) and validation (20%) sets, ensuring no overlap between individuals in the two subsets”, “In each iteration, the model performs uniform sampling from different datasets to form the training minibatch. We set the maximum number of training steps to 25,000,000, with early stopping based on validation performance every 2500 steps.”

We argue that the observed performance difference likely arises from the limited dataset diversity relative to the model’s increased parameter capacity.

Main Page 6 “Over the five scenarios examined, in certain cases, CSFM-Large, where applicable, occasionally exhibits slightly inferior performance compared to CSFM-Base. This observation suggests that its current dataset may be relatively insufficient to fully leverage the capacity of a larger model, in contrast to some existing large pretrained vision or language foundation models (e.g., GPT4), which benefit from extensive pretraining on vast datasets. Future research may investigate the scaling laws of training foundation models in the cardiac biosignal domain to optimize the balance between model capacity and available data.”

Future extensions could explore scaling pretraining through dataset expansions, such as incorporating data sources used by ECGFounder [4], which may further clarify the relationship between data diversity

and model capacity in this domain.

Nevertheless, in line with the computational cost discussion in C.4.4, these findings suggest that extremely large models may not be strictly necessary, both in terms of performance and efficiency, to achieve strong generalization across cardiac sensing tasks, although this remains an open question for future investigation.

C 4.4 — The model has large set of parameters, but limited analysis of inference time or computational requirements for clinical deployment.

Reply: Many thanks for this insightful point. We have made revisions as below,

Table 10: **Details of our cardiac sensing foundation models.** Att.: Attention; Enc.: Encoder. FLOPs are measured using a single-lead 10-second (250Hz) input segment, and also reported relative to ResNet1d18.

Model scale	# Parameters	Representation size		Transformer block		FLOPs (\times ResNet1d18)
		Hidden	Intermediate	Att. head	#Enc. layer	
CSFM-Tiny	51M	1024	768	8	6	0.86G (4.4x)
CSFM-Base	117M	3072	768	12	12	3.83G (19.9x)
CSFM-Large	343M	4096	1024	24	16	61.07G(70.6x)

a. In the revision, we have now added a detailed analysis in the Supplementary Material, as below,

Supplementary S3.7 “The details of CSFM parameters are provided in Table 10. We acknowledge that the computational cost is non-negligible for foundation models per se, which represents a limitation of the current implementation.

The CSFM family ranges from the relatively lightweight *CSFM-Tiny* to the full *CSFM-Large*, maintaining a balanced trade-off between parameter count, computational efficiency, and performance.

Furthermore, in the context of VTaC ICU deterioration events detection, which necessitate real-time prediction, CSFM demonstrates strong predictive performance (AUC over 0.7) even five minutes before alarm onset, as shown in Figure 4a of the main text. This highlights its potential applicability in real-time patient monitoring scenarios.

Nevertheless, optimizing CSFM for low-latency and resource-constrained environments remains an important direction for future work, particularly for deployment on wearable or embedded healthcare devices.”

b. We emphasize that efficiency remains a limitation, and we highlight the development of more efficient or lightweight adaptations of CSFM as an important direction for future work to further improve deployability, in the main text.

Main Page 10 “Finally, the computational cost associated with training and deploying large-scale transformer architectures is non-trivial, potentially limiting accessibility in resource-constrained settings.”

C 4.5 — There is significant performance differences between real and synthetic data (Table 3, suggesting quality gaps of synthetic data and real data.

Reply: Many thanks for highlighting this important point. In line with C 1.13, we acknowledge that there are significant performance differences between real and synthetic data, reflecting quality gaps between the two domains. We also made revisions in Supplementary Section S3.6 as below, about the potential mismatch reasons.

a. Diversity mismatch.

“Although CinC17 and SimBand have similar segment sizes of around 8k, CinC17 contains 8,528 subjects, whereas SimBand includes only 40 individuals, thus constraining the diversity and representativeness of the synthetic domain. ”

b. Data quality mismatch. “PPG signals (and occasionally ECG) are inherently noisy, and the use of a one-to-one reconstruction objective, such as mean squared error (MSE), can lead the model to learn overly smoothed waveforms that fail to capture real-world artifacts and the diverse physiological

associations between PPG and ECG. Previous findings in the computer vision domain [14] have shown that MSE-based generative optimization tends to produce over-smoothed results with limited diversity, motivating the development of more advanced generative approaches such as adversarial, diffusion, or variational models.”

Representative examples are shown in Figure 12b, where the generated ECG signals appear smoother than the real ECG traces, which contain more high-frequency fluctuations and physiological noise.

“As a result of these, training on synthetic signals may fail to capture the variability and noise encountered in real-world settings. This issue is particularly relevant for wearable datasets such as CinC17 and SimBand, where motion artifacts and device-specific noise play an important role in signal quality. As a result, models trained on CinC17 can generalize reasonably well to synthetic data, but models trained only on synthetic data fail to generalize effectively to CinC17.”

To mitigate the gap to some extent, we introduced specific data augmentation strategies during training, as reported in the Supplementary, which led to measurable performance gains.

“We introduced data augmentation strategies during synthetic pretraining, specifically temporal dropout, random scaling, and Gaussian noise, as shown in Figure 12a. These augmentations yielded measurable improvements and partially reduced the performance gap, although some of these gains may stem from the augmentations themselves.”

We believe there remains significant room for improvement, particularly through the integration of diversity-enhancing generative mechanisms, such as variational methods or diffusion-based models. We highlight this as a promising direction for future work.

Figure 12: **Cross-modality reconstruction and augmentation results.** **a.** PPG to ECG Reconstruction. (1) The reconstruction was performed on VitalDB, and the waveform reconstruction performance was reported on the held-out test set of VitalDB. (2) Subsequently, we applied the adapted model to the original SimBand dataset to generate synthetic Lead-II ECG waveforms. To comprehensively test the quality of generated ECG waveforms, we conducted two experimental settings: train on synthetic ECG from SimBand (normal versus AF), and test on real ECG on CinC17 (normal versus AF), and vice versa, with an independent model ResNet1d18. The performance is reported using F1 and AUC. **Best** values are in bold, and second best are underlined. **b.** Examples of real and synthetic ECG examples belonging to normal and AF.

C 4.6 — When evaluating the models, the paper sometimes used metrics which are not very clear: e.g., modified Macro-F1

Reply: Thanks, we have accordingly made revision in terms of the following two aspects

- a. In terms of the specific modified macro-F1, we have changed to “Performance was measured using the macro-F1 score, computed over only the valid answers only as a modified macro-F1 score.”
- b. We further added the illustration of the metrics in the supplementary and refer to this in the main text.

Supplementary Section S1.4 “The evaluation metrics for the three groups of downstream tasks are summarized as follows.

Binary classification. For binary tasks, the primary evaluation metric was the area under the receiver operating characteristic curve (AUC), as it provides a threshold-independent measure of discriminative performance.

Multi-class and multi-label classification. For multi-class and multi-label tasks, the primary metric was the macro-averaged F1-score, which equally weights all classes and is less affected by label imbalance. For completeness, Matthews correlation coefficient (MCC) values are also reported in the Supplementary Table 8 for complementary evaluation. In particular, the ECG-QA task is formulated as a multi-label classification problem, where the model selects the correct answer(s) from a predefined set of candidate options. When computing the metrics, false positives outside the given candidates are not considered. Therefore, a modified version of the F1-score [13] was used to better reflect task-specific accuracy.

Regression. For regression tasks, such as vital sign estimation, the mean absolute error (MAE) was used as the primary evaluation metric, as it is more interpretable and less sensitive to outliers than the root mean squared error (RMSE). The RMSE was additionally reported, in some cases, to provide complementary assessments.”

C 4.7 — There are also not much details about data preprocessing and limited details on signal preprocessing and quality control measures, e.g., “weak matching” between ECG signals and clinical notes in MIMIC-III-WDB could introduce noise in the multi-modal learning process.

Reply: Thank you very much for the critical points. We aimed to address the points in terms of the following,

a. Details in Supplementary.

We have expanded the Supplementary Materials to provide additional details on the preprocessing of text. For reproducibility, we have also put the preprocessing scripts used to refine the text inputs in the supplementary code.

b. Ablation study with/without “weak matching”.

We agree that “weak matching” between ECG signals and clinical notes can introduce noise. For example, in some cases, there may be imperfect alignment between the ECG trace recorded and the corresponding clinical note. To assess the impact of this, we conducted an ablation study comparing training with and without text input. The results show that although weak matching, such paired data does bring consistent performance gains for PPG compared to baselines.

The discussions and results are as below,

Supplementary Section S3.5 “Weak Match” facilitates ECG/PPG representation learning. On the other hand, it is noted that the weak matching approach, as mentioned in Section S1.3.3, may include information that is not immediately paired in the text and current biosignal snapshot. The results in Figure 13b compare pretraining on MIMIC-III PPG+ECG with that on MIMIC-III PPG+ECG+text; the latter achieves better or comparable performance across five tasks, which demonstrates the importance of such “weak” text associations. This approach is particularly valuable in real-world settings, where exact matching between text and biosignals is often not achievable.

Figure 13: **Ablation study for different pretraining settings.** **a.** Comparison of our training strategies against other “straightforward” solutions to handling heterogeneous health records, (i) keeping the common channels only, *i.e.*, Lead-II ECG, (ii) keeping one dataset only, *i.e.*, MIMIC-IV including both 12-Lead ECGs and texts. Based on our training strategies and these two compared strategies, we assessed the performance disparity across varied lead settings on PTB-XL datasets. The radar axes in the figure are log-normalized and use the same range for better visualization. **b.** Comparison of pretraining strategies under different lead modalities and dataset combinations. Ablation models pretrained with (i) MIMIC-III PPG-only, (ii) MIMIC-III PPG+ECG, (iii) MIMIC-III PPG+ECG+Text, and (iv) the full heterogeneous pretraining set were compared across five PPG-related tasks. Performance values were min-max normalized within each task for clearer comparison.

C 4.8 — Some figures have overlapping text(fig 8)

Reply: Thanks, it has been fixed. The updated figure (now Supplementary Figure S4a) is shown in Figure 13a.

C 4.9 — The code is organized well with clear structure and documentation of README with clear installation instructions and usage examples, the author also provided a tutorial notebook with step-by-step guidance and separated code part of model definition, data loading, and training. The only limitation is some parameters are hard coded and may limit the model flexibilities.

Reply: Many thanks, we have revised the code to remove hard-coded parameters and instead expose them as configurable arguments.

Users can now initialize the model flexibly either by manually specifying arguments or by selecting predefined variants. Updated codebase are in the supplementary materials.

Before (hard-coded initialization)

```
model = CSFM(  
signal_size=2500, patch_size=25, num_classes=1, channels=13,  
dim=768, depth=6, heads=8, mlp_dim=1024, dropout=0.1, emb_dropout=0.1,  
text_len=64, pool='cls')
```

After (soft-coded, configurable)

```
model = CSFM_model('Tiny')  
model = CSFM_model('Base')  
model = CSFM_model('Large')
```

References

- [1] Y. Luo, Y. Chen, A. Salekin, and T. Rahman, “Toward foundation model for multivariate wearable sensing of physiological signals,” 2024. [Online]. Available: <https://arxiv.org/abs/2412.09758>
- [2] A. F. Ansari, L. Stella, A. C. Turkmen, X. Zhang, P. Mercado, H. Shen, O. Shchur, S. S. Rangapuram, S. P. Arango, S. Kapoor *et al.*, “Chronos: Learning the language of time series,” *Transactions on Machine Learning Research*.
- [3] M. Goswami, K. Szafer, A. Choudhry, Y. Cai, S. Li, and A. Dubrawski, “Moment: A family of open time-series foundation models,” in *International Conference on Machine Learning*. PMLR, 2024, pp. 16 115–16 152.
- [4] J. Li, A. D. Aguirre, V. M. Junior, J. Jin, C. Liu, L. Zhong, C. Sun, G. Clifford, M. Brandon Westover, and S. Hong, “An electrocardiogram foundation model built on over 10 million recordings,” *NEJM AI*, vol. 2, no. 7, p. AIoa2401033, 2025.
- [5] M. P. Hung, A. Saeed, and D. Ma, “Boosting masked ecg-text auto-encoders as discriminative learners,” in *Forty-second International Conference on Machine Learning*.
- [6] C. Liu, Z. Wan, C. Ouyang, A. Shah, W. Bai, and R. Arcucci, “Zero-shot ecg classification with multimodal learning and test-time clinical knowledge enhancement,” in *International Conference on Machine Learning*. PMLR, 2024, pp. 31 949–31 963.
- [7] A. Pillai, D. Spathis, F. Kawsar, and M. Malekzadeh, “Papagei: Open foundation models for optical physiological signals,” *arXiv preprint arXiv:2410.20542*, 2024.
- [8] K. McKeen, L. Oliva, S. Masood, A. Toma, B. Rubin, and B. Wang, “Ecg-fm: An open electrocardiogram foundation model,” *arXiv preprint arXiv:2408.05178*, 2024.
- [9] G.-Q. Zhang, L. Cui, R. Mueller, S. Tao, M. Kim, M. Rueschman, S. Mariani, D. Mobley, and S. Redline, “The national sleep research resource: towards a sleep data commons,” *Journal of the American Medical Informatics Association*, vol. 25, no. 10, pp. 1351–1358, 2018.
- [10] L.-w. Lehman, B. Moody, H. Deep, F. Wu, H. Saeed, L. McCullum, D. Perry, T. Struja, Q. Li, G. Clifford *et al.*, “Vtac: a benchmark dataset of ventricular tachycardia alarms from icu monitors,” *Advances in Neural Information Processing Systems*, vol. 36, 2024.
- [11] E. M. Lima, A. H. Ribeiro, G. M. Paixão, M. H. Ribeiro, M. M. Pinto-Filho, P. R. Gomes, D. M. Oliveira, E. C. Sabino, B. B. Duncan, L. Giatti *et al.*, “Deep neural network-estimated electrocardiographic age as a mortality predictor,” *Nature communications*, vol. 12, no. 1, p. 5117, 2021.
- [12] P. Wagner, N. Strodthoff, R.-D. Bousseljot, D. Kreiseler, F. I. Lunze, W. Samek, and T. Schaeffter, “Ptbx-xl, a large publicly available electrocardiography dataset,” *Scientific data*, vol. 7, no. 1, pp. 1–15, 2020.
- [13] J. Oh, G. Lee, S. Bae, J.-m. Kwon, and E. Choi, “Ecg-qa: A comprehensive question answering dataset combined with electrocardiogram,” in *Advances in Neural*

Information Processing Systems, A. Oh, T. Neumann, A. Globerson, K. Saenko, M. Hardt, and S. Levine, Eds., vol. 36. Curran Associates, Inc., 2023, pp. 66 277–66 288. [Online]. Available: https://proceedings.neurips.cc/paper_files/paper/2023/file/d0b67349dd16b83b2cf6167fb4e2be50-Paper-Datasets_and_Benchmarks.pdf

- [14] C. Ledig, L. Theis, F. Huszár, J. Caballero, A. Cunningham, A. Acosta, A. Aitken, A. Tejani, J. Totz, Z. Wang *et al.*, “Photo-realistic single image super-resolution using a generative adversarial network,” in *Proceedings of the IEEE conference on computer vision and pattern recognition*, 2017, pp. 4681–4690.
- [15] X. Gu, Y. Guo, F. Deligianni, B. Lo, and G.-Z. Yang, “Cross-subject and cross-modal transfer for generalized abnormal gait pattern recognition,” *IEEE Transactions on Neural Networks and Learning Systems*, vol. 32, no. 2, pp. 546–560, 2020.

Response to review comments

NATMACHINTELL-A25073100-B

We thank the reviewers and the editors for their careful reading of our manuscript and for the positive feedback, which has indeed helped to further improve this work during the last round.

We have revised the manuscript in accordance with the editorial requirements for the final version and have carefully checked the text/figures to correct any typographical errors. We sincerely appreciate the opportunity to submit the final version.